# Epithelial cells maintain memory of prior infection with *Streptococcus pneumoniae* through di-methylation of histone H3

Christine Chevalier [1], Claudia Chica[2], Justine Matheau [1,3], Adrien Pain [2], Michael G. Connor[1] & Melanie A. Hamon [1]✉

Epithelial cells are the first point of contact for bacteria entering the respiratory tract. *Streptococcus pneumoniae* is an obligate human pathobiont of the nasal mucosa, carried asymptomatically but also the cause of severe pneumoniae. The role of the epithelium in maintaining homeostatic interactions or mounting an inflammatory response to invasive *S. pneumoniae* is currently poorly understood. However, studies have shown that chromatin modifications, at the histone level, induced by bacterial pathogens interfere with the host transcriptional program and promote infection. Here, we uncover a histone modification induced by *S. pneumoniae* infection maintained for at least 9 days upon clearance of bacteria with antibiotics. Di-methylation of histone H3 on lysine 4 (H3K4me2) is induced in an active manner by bacterial attachment to host cells. We show that infection establishes a unique epigenetic program affecting the transcriptional response of epithelial cells, rendering them more permissive upon secondary infection. Our results establish H3K4me2 as a unique modification induced by infection, distinct from H3K4me3 or me1, which localizes to enhancer regions genome-wide. Therefore, this study reveals evidence that bacterial infection leaves a memory in epithelial cells after bacterial clearance, in an epigenomic mark, thereby altering cellular responses to subsequent infections and promoting infection.

The external and internal surfaces of the body are covered with epithelia, which form a barrier to the environment. As such, epithelial barriers and epithelial cells are the primary responders to environmental assault and constitute the first line of defense against invading pathogens. They play an important role in controlling the initial steps in the inflammatory response upon contact with virulent pathogens while maintaining tissue homeostasis under resting state conditions[1,2]. Therefore, tight regulation of epithelial cell responses is paramount to ensure homeostatic maintenance even under contact with harmless bacteria and response to danger signals.

Pathogenic bacteria are sensed by the epithelia, mainly through pattern recognition receptors, but also have evolved mechanisms to subvert host responses to impose infection. One of these strategies involves reprogramming host transcription, effectively down-regulating inflammatory responses or promoting cell survival. There are multiple levels at which bacteria can modulate host transcription; at the level of signal transduction, transcription factor regulation, or even at the level of chromatin organization[3–5]. Eukaryotic DNA is organized in a complex structure combining DNA and associated proteins into chromatin. The primary unit of chromatin, the nucleosome, is composed of an octamer of 4 core histones, H2A, H2B, H3, and

[1]Institut Pasteur, Université Paris Cité, Chromatin and Infection Laboratory, F-75015 Paris, France. [2]Institut Pasteur, Université Paris Cité, Bioinformatics and Biostatistics Hub, F-75015 Paris, France. [3]Bio Sorbonne Paris Cité doctoral school, Department of Infectiology Microbiology, Université Paris Cité, F-75006 Paris, France. ✉e-mail: melanie.hamon@pasteur.fr

H4, around which DNA is wrapped. The position of nucleosomes, as well as the extent to which DNA is wrapped, plays an essential role in the accessibility of genetic regions to transcription factor binding and transcription machinery, and thereby regulates transcription. The chromatin structure is dynamically regulated at the nucleosome level by post-translational modifications of histone proteins as well as remodeling enzymes that reposition nucleosomes along the genome. Interestingly, bacterial pathogens have been shown to actively induce chromatin modifications as a potent transcriptional reprograming strategy[4,6,7].

*Streptococcus pneumoniae* is one example of a bacterium that actively drives chromatin remodeling. *S. pneumoniae* (or pneumococcus) is a natural colonizer of the human upper respiratory tract, which is prevalently carried asymptomatically in a large proportion of the population[8]. However, it is the cause of the largest number of community-acquired pneumonia cases[9], making it also a deadly pathogen. Considering that vaccine evasion is prominent, and that antibiotic resistance is continuously increasing, the pneumococcus is on the WHO list of priority pathogens[10]. This bacterium has long been defined by its capsule, which has been used to classify *S. pneumoniae* into different serotypes, and against which the vaccine is directed. However, genome-wide analysis of over 2000 strains has revealed that, although serotype is a potent virulence determinant, genomic variation beyond serotype largely contributes to virulence[11]. Strikingly, pneumococcus has been shown to actively modify host chromatin to drive infection, but also to maintain homeostatic colonizing conditions, depending on the bacterial strain[12,13]. In fact, histone modifications induced by pneumococcus have been shown to be mediated by bacterial proteins rather than capsule[12,13].

Chromatin modifications in an inflammatory setting have been documented for many years. For instance, stimulation of macrophages with the bacterial outer membrane component lipopolysaccharide (LPS) causes chromatin remodeling at numerous inflammatory genes[14]. Although many of these modifications are transient and return to basal state once the stimulus has been removed, in some instances, stimuli induce lasting modifications that alter the transcriptional response of the cell. This is best exemplified in innate immune memory (also termed "trained immunity"), in which innate immune cells exposed to a particular stimulus (infectious agent, inflammatory signals, etc.) mount a greater and faster response against secondary challenge, whether it is homologous or heterologous[15]. Although still the subject of many investigations, chromatin remodeling is attributed to maintaining such memory responses. Indeed, immune memory has been associated with histone modification at enhancer regions, transcription of long non-coding RNAs (lncRNAs), DNA methylation, and reprograming of cellular metabolism[16]. Most of the studies of immune memory have concentrated on cells of the immune system, such as monocytes, macrophages, or NK cells, but little has been done on epithelial cells. Only recent studies have emerged documenting memory responses in these cells important for barrier functions[17,18]. Immune memory in different settings is, therefore, an important host response for protection against infection. However, how bacteria impact immune memory, or whether bacteria-induced histone modifications impose their own memory has not been explored.

In this study, we demonstrate that *S. pneumoniae* infection induces a persistent histone modification, di-methylation of histone H3 on lysine 4 (H3K4me2), which is maintained at least 9 days upon clearance of bacteria with antibiotics. Our results establish H3K4me2 as a unique modification induced by infection, distinct from H3K4me3 or H3K4me1, which localizes to enhancer regions genome-wide. We show that following infection a unique epigenetic program is acquired which is associated with modifications of the transcriptional response and the function of epithelial cells. Importantly, epithelial cells are changed following infection and respond differently upon a secondary infection many days after, displaying alteration in their metabolism, and lysosomal transport. This process is actively driven by bacteria and favors

greater bacterial adhesion upon secondary infection. Therefore, H3K4me2 is associated with the memory of passed infections in epithelial cells and with altered cellular responses favoring infection.

## Results

### Cells respond differently in primary and secondary infections

We hypothesized that epithelial cells could retain the memory of a first infection, and this would result in a different response upon secondary infection. To test this, we set up an infection model in which alveolar epithelial lung A549 cells were infected with TIGR4 *Streptococcus pneumoniae* strain (~$1 \times 10^8$ CFU/mL) or not (UI) (Fig. 1A). After infection, cells are washed, and antibiotics are added to the medium. Importantly, no more CFUs were recovered 24 hours post infection. The cells are infected once (primary infection or 1°), twice (secondary infection or 2°), or are maintained in culture for several cell divisions following a first infection (PI). For cells maintained in PI and 2° conditions, two cell passages are performed following a first infection to ensure that cells exposed to bacteria a second time are not the same as those exposed the first time, but instead are descendants. Samples are collected throughout this study at different times indicated by α, β, γ. We assessed cell viability using live and dead cell dyes (Calcein AM and DRAQ7™, respectively) following primary and secondary infections (samples collected at α and γ) in comparison to uninfected cells (Fig. S1A). Live cells are measured in quadrant Q3, while dead cells in Q1. For every time point monitored, >90% of cells remain alive. Furthermore, we controlled that cell proliferation is not affected after 1° infection by staining cells with the CellTrace™ CFSE dye (Fig. S1B). Therefore, our model accurately measures the persistent effect of infection on live cells without bias for subpopulations of resistant cells.

Using this protocol, we performed transcriptomic analysis to compare host cell expression under the four conditions, UI, 1°, PI and 2°. Principal component analysis (PCA) of the results shows that the main source of variability (PCA1 ~ 50%) reflects the differences between active infection (1° and 2°) versus the lack of bacterial presence (UI and PI) (Fig. 1B). Interestingly, PC2 (~11%) reflects the differential response between the two infections and no variability between UI and PI conditions is observed. Differentially expressed genes (DEGs, adjusted *p* value < 0.05) were determined for the 1° and 2° infections (1° vs UI and 2° vs PI) (Fig. 1C). In total, there are 1310 upregulated and 1491 downregulated genes. Of which, 560 upregulated and 593 downregulated are significantly different only in the 1° infection in comparison to uninfected cells (green bars), and 383 upregulated and 490 downregulated are specific of the 2° infection. These results show that, even if a comparable number of genes are differentially upregulated or downregulated, <30% of these changes are shared between the 1° and the 2° infection (367/1310 for upregulated genes and 408/1491 for downregulated). This highlights the observation that cells following 1° infection respond differently to a 2° infection.

By performing Gene Set Enrichment Analysis (GSEA), we obtained a functional description of the transcriptome using the gene sets reported in the Reactome database (Fig. 1D and Table S1). A normalized enrichment score (NES) is calculated for each significatively enriched pathway (FDR < 0.05). Interestingly, some of the functional categories identified are different between the two infections, and some are common. A heatmap representation of the NES per functional category is shown in Fig. 1D, and a graphical representation of the expression fold change per category is shown in Fig. S1C. Figure 1D clearly shows that pathways specific to 1° infection show slighter differences in NES than those specific to 2° infection, indicating that new gene categories become significant in 2° infection. It should be noted that although some functional categories are shared between infections, they may include different pathways in each infection (Table S1). Remarkably, differential gene expression between 1° and 2° infection is specific to certain categories of genes, including chromatin organization as specific to 1° infection, metabolism of proteins, and cellular

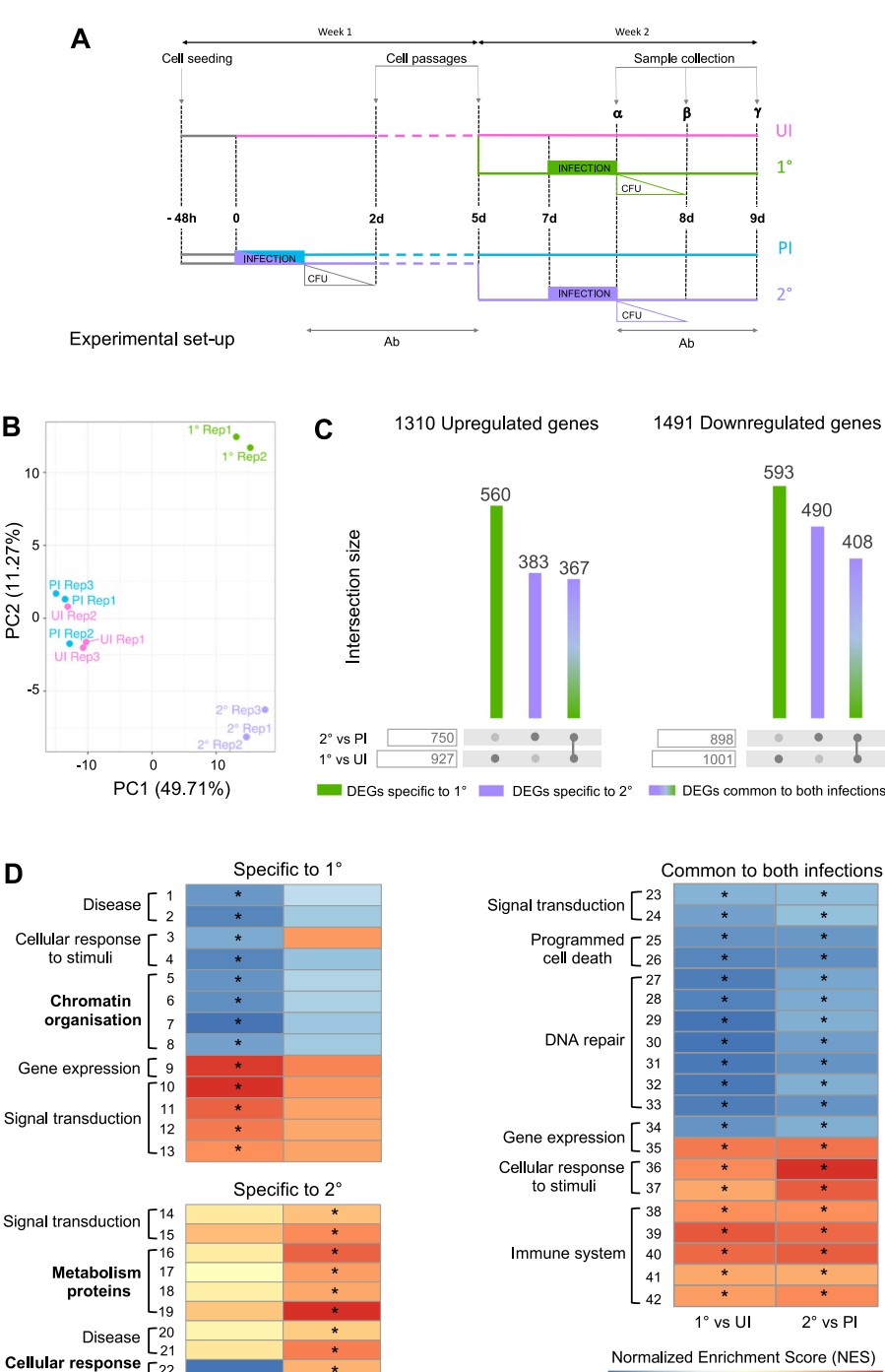

**Fig. 1 | Differential infection efficiency and host cell transcriptome between primary and secondary infections. A** Experimental scheme, showing A549 or RPMI cells infected with *S. pneumoniae*, pink for uninfected cells (UI), green for primary infection (1°), blue for cells maintained post first infection (PI), purple for secondary infection (2°). Addition of an antibiotics cocktail (Ab) 1 or 3 hours after infection is indicated below the scheme and correlates with the rapid decrease in colony forming units (CFUs) after infection. Sample collection times are indicated by: $\alpha = 1$ h or 3 h (for 1° and 2°) and 7d for (PI), $\beta = 24$ h (for 1° and 2°) and 8d (for PI), $\gamma = 48$ h (for 1° and 2°) and 9d (for PI). **B** Principal component analysis of gene expression A549 cells of the UI, PI, 1° and 2° infection at time α (3 h). Principal components calculated on count data after removal of replicate batch effect with ComBat. Samples are colored according to the time point, all replicates (Rep) are shown. **C** Differential expression analysis. Number of genes significantly (adjusted *p* value < 0.05) up or downregulated in the 1° or 2° infection in comparison, respectively, to UI and PI cells. Column height indicates the number of genes defined as differentially expressed (DEG) in one or more of the above comparisons. Column color specifies whether the subset of genes is differentially expressed only in the 1° (green) or 2° (purple) infection or in both (green/purple gradient). **D** Functional scoring analysis. Reactome categories, selected among those significantly enriched (False discovery rate, FDR < 0.05) in the 1° and/or 2° infection in comparison to, respectively, UI and PI cells are indicated by star. Heatmaps represent the intensity of the Normalized Enrichment Score (NES) obtained by the Gene Set Enrichment Analysis (GSEA) for each pathway. Pathways are separated depending on whether they are enriched specifically in the 1° or 2° infection, or in both. Categories of genes with differential gene expression between 1° and 2° infection are in bold. Complete list of pathways corresponding to each category in supplementary table S1. Source Data were deposited into the Gene Expression Omnibus (GEO) repository of the National Center for Biotechnology Information under accession number GSE230142.

response to stimuli as specific to 2° infection, whereas no significant differences in cytokine or antibacterial molecules were observed either by transcriptome analysis or by ELISA (Fig. S1D). Overall, our results reveal that the transcriptional status of cells upon a 2° infection is specifically modified on a subset of genes in comparison to 1°, suggesting a transcriptional memory is induced following 1° infection, which does not include antibacterial responses.

### *S. pneumoniae* actively modifies cells following a primary infection

Given that the transcriptional status of cells is affected following a primary infection, we further tested several infection-related phenotypes to establish the impact of infection. With the same protocol defined in Fig. 1A, we evaluated the infection efficiency during the primary (1°) and secondary (2°) infection. $3 \times 10^4$ bacteria are recovered on average from a 1° infection. It should be noted that as *S. pneumoniae* is mainly extracellular, with a minor proportion of bacteria entering cells, the CFU counts recovered after infection include both extracellular as well as some intracellular bacteria. Strikingly, approximately 6-fold more bacteria are recovered following a 2° infection, pointing to a greater permissivity of cells upon secondary exposure (Fig. 2A). We further determined that the higher level of recovered CFUs were bacteria tightly adhering to cells, rather than cells in the medium. Moreover, the increase in bacterial numbers is only observed following primary infection with live bacteria, as similar numbers to those obtained in 1° are recovered post incubation with inactivated pneumococcus (Fig. 2B). Therefore, *S. pneumoniae* actively modifies cells rendering them more permissive upon secondary exposure. It is interesting to note that even though there are 6 times more bacteria upon 2° infection, antibacterial response genes are not more highly activated than in 1° infection, arguing that cells do not display innate immune memory of prior infections (Fig. S1D).

Metabolism of host cells is often affected upon infection. Thus, we monitored the metabolic activity of cells using Resazurin, an oxidation-reduction indicator that becomes Resorufin, fluorescent upon reduction in the reducing environment of metabolically active cells (Fig. 2C). Interestingly, Resorufin fluorescence increased at a significantly faster rate following 1° infection compared to 2° infection, demonstrating a slower oxidation-reduction rate during 2° infection. Similarly to what is observed for bacterial recovery, the loss in metabolic activity during 2° infection is only observed if the 1° infection was performed with live bacteria.

As lysosomes are key metabolic signaling centers[19], we further monitored their level and acidity following 1° and 2° infections. Quantification of lysosomes was performed using LAMP1 staining of fixed cells. Although 1° infection leads to a small increase in LAMP1 staining intensity, 2° infection increases it even more (Fig. 2D and S2A). Interestingly, this effect is only observed at 3 h post infection, when bacteria are present and alive, whereas no difference is observed at 24 h, post antibiotic treatment. In addition, a significant increase in the acidity of lysosomes, as measured using lysotracker, was also observed specifically during 2° infection. Surprisingly, bacteria and LAMP1 do not colocalize in either 1° or 2° infections (Fig. S2B), suggesting that lysosomes are not filled with bacterial products. These observations further confirm that the increase in bacterial numbers upon 2° infection reflects an increase in adhesion and not invasion. Therefore, endosomal trafficking is also modified following 1° infection.

It is interesting to note that modifications following 1° infection only affect certain pathways. Indeed, DNA damage was measured by monitoring the level of γH2Ax, which is essential to the efficient recognition and/or repair of DNA double-strand breaks (Fig. S2C). Although infection clearly induces a measurable level of DNA damage, these levels are similar during 1° and 2° infections.

### Cells maintain an epigenetic mark after infection

Our data demonstrate that 1° infection changes the cells in a lasting manner, suggesting a mechanism to maintain a memory of infection through multiple cell divisions. This raised the question of whether an epigenetic mechanism could contribute to these changes, with a particular interest in histone modifications. To identify potential modifications, we performed a multiplex ELISA, which analyzed twenty-one modified histone H3 patterns simultaneously in histone extracts from cells at 3 hours (1°) and 7 days (PI) after infection compared to uninfected cells (UI) (Fig. 3A and S3A). Interestingly only one histone modification was significantly detected at 7 days following bacteria challenge, di-methylation of histone H3 on lysine 4 (H3K4me2). This modification is not reliably detected directly after 1° infection and appears later. We further measured H3K4me2 levels in whole cell extracts by immunoblotting (Fig. 3B and S3B). By comparing the level of modified H3, we confirmed the significant increase at 7 days (PI) compared to 3 h (1°) and uninfected cells (UI). Given that H3K4me2 is often associated with H3K4me3, as the methyltransferase is the same, we also measured levels of tri-methylation under the same conditions. However, the level of H3K4me3 does not change either upon 3 h of infection (1°) or 7 days after infection (Fig. 3C and S3C). With this data, we concluded that infection leaves one specific epigenomic mark after bacterial clearance and that it persists for at least 7 days.

To further assess the specificity of this lasting modification, we measured the levels of H3 phosphorylation, as we had previously shown that *Sp* infection induces dephosphorylation on serine 10[13]. Although a significant decrease in phosphorylation levels is observed 3 h post 1° infection, the levels returned to uninfected levels by 7 days (PI) (Fig. 3D and S3D). Therefore, H3S10 dephosphorylation is transient in comparison to H3K4me2, which is maintained.

In vivo relevance of this modification was also monitored following intranasal inoculation in a mouse model (Fig. 3E; The mouse was created with BioRender.com released under a Creative Commons Attribution-Noncommercial-NoDerivs Academic license, agreement number: PM26VA8SLI). 3 days post infection, lungs were collected, and the level of H3K4me2 was measured in epithelial cells by FACS analysis. Strikingly, a consistent increase in the levels of di-methylation was observed upon infection as compared to a PBS control (Fig. 3F). Therefore, H3K4me2 is a chromatin mark maintained after infection with *S. pneumoniae* both in vitro and in an in vivo physiological model of infection.

### H3K4me2 levels are actively induced by live bacteria and dependent on cellular binding

We evaluated how the increase in di-methylation levels of H3K4 is distributed from cell to cell by performing immunofluorescence analysis to quantify the level of histone modification in individual cells. Images show a slight variability of H3K4me2 levels from one cell to another, with some cells staining more intensely than others under all conditions (Fig. 4A). Greater magnification reveals di-methylation of H3K4 is localized to euchromatin under all conditions, which is where most transcriptional activity takes place (Fig. S4A). Images were quantified for many nuclei to evaluate the levels of H3K4me2 in Fig. 4B. The global levels significantly increased at 48 hours following 1° infection and were still elevated 9 days (PI). Images show that both the number and the intensity of fluorescence increase in both conditions. This increased level of H3K4 di-methylation 9 days after infection (PI live) does not change after a second infection (2°) (Fig. S4B), suggesting that these levels are maximal 9 days post infection and cannot be increased further by subsequent infections.

Given that pneumococci are mainly carried in the upper respiratory tract, and that A549 cells are type II pneumocytes, we further measured H3K4me2 levels in nasal RPMI 2650 cells. Similarly to A549 cells, we observe a strong increase in the level of H3K4me2 after 24 and 48 hours of infection (1°) (MOI 10) compared to uninfected cells (UI)

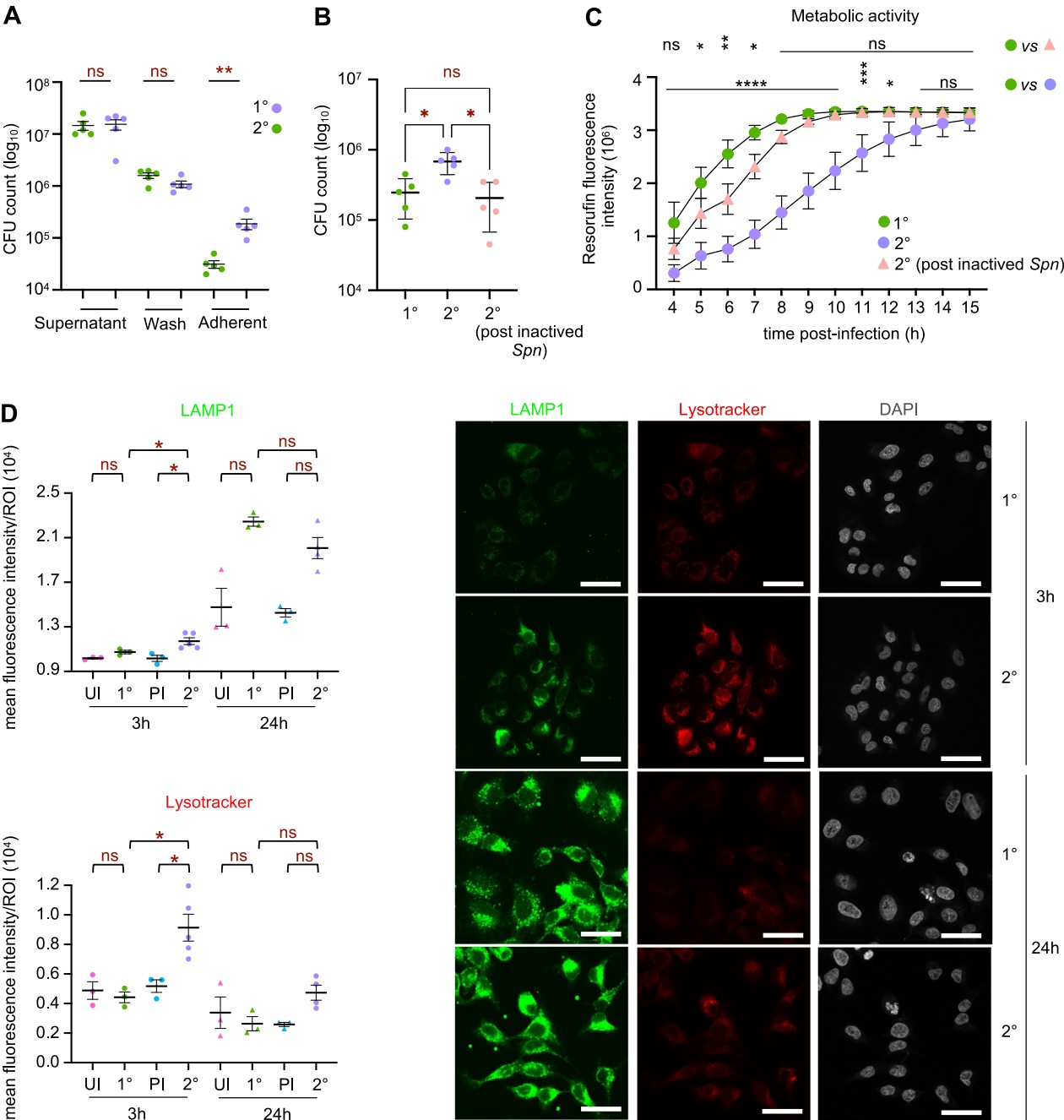

**Fig. 2 | *S. pneumoniae* actively modifies cells following primary infection.**
**A** Supernatant, wash, and A549 cells were collected at time α (1 h) post 1° and 2° infection with MOI 35 for CFU counts, $n = 5$ biological replicates, lines are the mean ± SEM and statistical significance was calculated by Mann-Whitney test, ns = not significant, two-tailed **$p = 0.0079$. **B** A549 cells were collected at time α (3 h) after 1° and 2° (MOI 20) for CFU counts, $n = 5$ biological replicates, lines are the mean ± SEM and statistical significance was calculated by two-tailed Mann-Whitney test, ns = not significant, *p = 0,0159 (1°/2°), *p = 0,0238 (2°/2°post inactived *Spn*). **C** A549 cell metabolic activity measured by alamarBlue assay. Cells treated at time α (3 h post infection, MOI 20), 1 measure of fluorescence/hour for 12 hours. Results are expressed as resorufin fluorescence intensity. Plot shows mean ± SEM from $n = 4$ biological replicates. Statistical significance was determined by two-way ANOVA with FDR Benjamini-Hochberg correction for multiple comparisons

Q = 0.05 for 1° (green round) versus 2° infection (round purple) ns = not significant, *$p = 0.0145$, ***$p = 0.0004$, ****$p = 0.0001$ and 1° (green round) versus 2°§ infection (§*Spn* inactivated at 1° infection then *Spn* live at 2°, triangle pink) ns = not significant, *$p = 0.0311$ and 0.0170, **$p = 0.0021$. **D** Quantification of LAMP1 and colocalization with lysotracker® at time α (3 h, round) and time β (24 h, triangle) for UI, 1°, PI and 2° conditions in the A549 cells. Plots show quantifications from 60 to 75 cells by conditions in the same experiment, mean fluorescence intensity of Regions of Interest (ROI) from $n$ = between 6000 and 22,000 lysosomes total (Fig. S2.B) ± SEM, statistical significance was determined by two-tailed Mann–Whitney test, ns = not significant, *$p = 0.0357$. On the right, representative immunofluorescence images by confocal microscopy of A549 cells stained with LAMP1 (GFP; green), Lyso-Tracker® (Deep Red; red), and DAPI (gray) at time α (3 h) and β (24 h) after 1° and 2° infection. Scale bar = 20 μm. Source data are provided as a Source Data file for **A**–**D**.

(Fig. 4C). In these cells, we also confirmed the specificity of our antibody by performing the same immunofluorescence experiments with another H3K4me2 antibody cataloged in the Histone Specificity Database (http://www.histoneantibodies.com). Similarly, we observe a

strong increase in the level of H3K4me2 after 48 hours of infection (1°) compared to uninfected cells (UI) (Fig. S4C), independently of the antibody used. Together these results show that H3K4 is di-methylated by infection in multiple epithelial cell types.

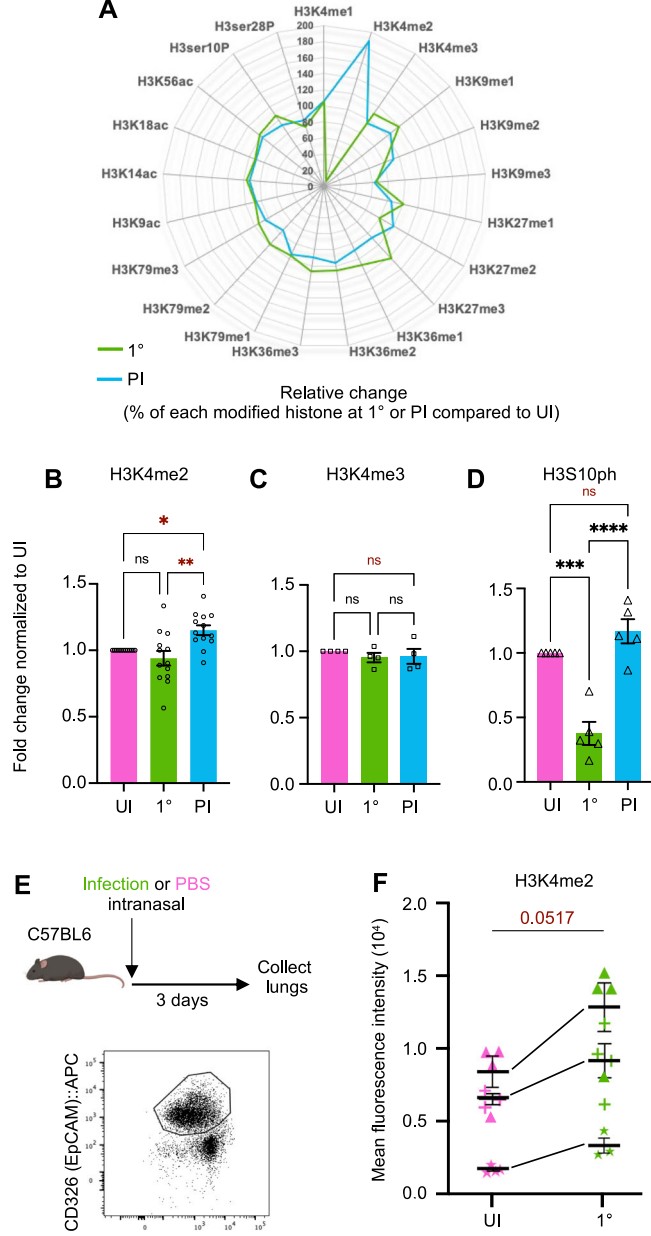

**Fig. 3 | Cells maintain an epigenetic mark after primary infection.**
**A** Quantification of twenty-one modified histone H3 patterns by Multiplex ELISA assay in the A549 cells. Radar plots representing the relative change (%) of each histone H3 modification between 1° or PI and UI cells at time α (3 h). Values in Supplementary Fig. S3.A. **B** Immunoblot detection of H3K4me2 at 1°, PI and UI at time α (3 h). Mean ± SEM of values expressed as normalized band intensity relative to β-actin followed by fold change of infected cells at 1° or PI to UI, n = 13 biological replicates. Statistical significance was determined by one-way ANOVA method with Tukey's multiple comparisons test (ns = not significant, *p = 0.0209, **p = 0.0011). Representative image of immunoblot in Supplementary Fig. S3.B **C** Immunoblot detection of immunoblot in Supplementary Fig. S3.B **C** Immunoblot detection of H3K4me3 1°, PI and UI at time α (3 h). Data points represent mean ± SEM, n = 4 biological replicates. Histogram shows the values expressed as normalized band intensity relative to β-actin followed by fold change of infected cells at 1° or PI to UI. Statistical significance was determined by one-way ANOVA method

with Tukey's multiple comparisons test (ns = not significant). Representative image of immunoblot in Supplementary Fig. S3.C. **D** Immunoblot detection of H3S10ph at 1° or PI to UI at time α (3 h). Data points represent mean ± SEM, n = 5 biological replicates. Histogram shows values expressed as normalized band intensity relative to β-actin followed by fold change of infected cells at 1° or PI to UI. Statistical significance was determined by one-way ANOVA method with Tukey's multiple comparisons test (ns = not significant, ***p = 0.0002, ****p < 0.0001). Representative image of immunoblot Supplementary Fig. S3.D. **E** In vivo experimental set-up described in Methods section. Illustration of mouse created with BioRender.com. **F** Mean intensity levels of H3K4me2 of epithelial cells from mice treated with PBS (UI) or infected (1°) is represented by a symbol for each mouse, n = 3 biological replicates (stars, crosses, triangles) with 3 or 4 mice per condition, mean ± SEM for each biological replicate. Statistical significance was determined by ratio paired t test of mean (p = 0.0517). Source data are provided as a Source Data file for **A–F**.

To demonstrate that H3K4 di-methylation is actively induced during infection, we determined whether the increase in H3K4 di-methylation is proportional to the bacterial load. We performed infections with increasing numbers of bacteria and measured H3K4me2 levels by immunofluorescence (Fig. 4D). These results

clearly show that increasing the bacterial load increases the levels of H3K4me2. We further monitored H3K4me2 levels upon incubation of cells with inactivated bacteria (Fig. 4A, B). These results show that di-methylation is only increased following infection with live bacteria. Importantly, these data also indicate that the increase in bacterial load

and metabolic alterations reported in Fig. 2C are dependent on H3K4me2. Together, these results show that the H3K4me2 epigenetic modification of cells is necessary for seco ndary phenotypes and is a feature of a live infection rather than due to a cellular response to the passive presence of bacterial factors.

We further determined whether bacterial factors impacting bacterial adhesion were important for the induction of H3K4me2. We compared the levels of H3K4me2 induced by wild-type pneumococcus (+) to those induced by a mutant lacking capsule (-) (Fig. 4E). For both strains, the levels of H3K4me2 were significantly increased compared to uninfected conditions and were slightly higher in response to the capsule mutant. These results demonstrate that capsule is not required for the observed increase in H3K4me2 and are consistent with results showing a lack of increase with inactivated bacteria, which retain capsule. In addition, we tested the ability of a mutant lacking pilus to induce H3K4me2 (Fig. 4F). Interestingly, only the wild-type strain induced an increase in H3K4me2 levels, and the pilus mutant did not. Therefore, the pilus is necessary for the observed increase in H3K4me2. Together, these data show a direct correlation between the level of bacterial attachment and the increase in H3K4me2, as a capsule mutant, which binds to A549 cells more strongly, leads to higher levels, whereas a pilus mutant less able to bind, does not induce di-methylation of H3 (Fig. S4D, E). Therefore, bacterial binding drives H3 di-methylation.

### H3K4me2 is a specific persistent histone mark

As H3K4 can be mono-, di-, or tri-methylated, we assessed whether other modifications on this residue were also altered upon infection. We measured levels of H3K4me3 and me1 by immunofluorescence in the same manner we determined the levels of H3K4me2. Results in Fig. 5A, B show that although a transient increase in both me3 and me1 is observed following infection (24 h and 48 h, respectively), but do not persist for 7 days or 9 days, respectively. Thus, our data demonstrate that H3K4me2 is the only modified form of H3K4 that persists at least 9 days following infection with *Sp*.

To begin to understand how infection leads to increased levels of H3 di-methylation, we used the pan-methyltransferase inhibitor Sinefungin, which we added to cells directly after infection (Fig. 5C). Samples were collected 48 h later to assess the levels of H3K4me2. Surprisingly, none of the concentrations of inhibitor tested blocked infection-induced di-methylation (Fig. 5D), even though these concentrations adequately blocked mono-methylation (Fig. 5E). Similarly to di-methylation, tri-methylation remained unchanged in the presence of Sinefungin (Fig. 5F), strongly suggesting that bacterial infection is not increasing the activity of a host methyltransferases, rather it could be blocking a demethylase.

We have previously shown that *S. pneumoniae* induces a strong histone H3 dephosphorylation on serine 10 during infection through activation of the host PP1 phosphatase[13], which we hypothesized could be a precursor modification to H3K4me2. Therefore, we used Calyculin A, a PP1 inhibitor, to block H3S10 dephosphorylation and assessed the levels of H3K4me2 48 h later (Fig. 5G). Although the inhibitor was effective in blocking H3S10 dephosphorylation (Fig. S5D), it had no effect on the levels of H3K4me2, demonstrating that both modifications are independent.

### Methylome dynamics during 1° infection

To study the distribution of H3K4me2 along the host cell genome, we used chromatin immunoprecipitation (ChIP-seq) to profile the methylome upon infection. The analysis recovered ~28,000 peaks 7 days post infection (PI) but only ~12600 directly after infection (Fig. 6A), which is consistent with immunofluorescence experiments showing a steady increase in H3K4me2 levels following 1° infection until 7 days (PI). Interestingly most of the peaks recovered following 1° infection are shared peaks with both UI and PI conditions, and only ~1

700 (~13%) are unique to that condition, supporting the idea that this time point is transitional. In contrast, more than ~30% of peaks unique to the PI condition (8221) are unique and distinct from either UI or 1° conditions. These results are consistent with a principal component analysis along the 1° infection time course (Fig. 6B), which reveals that the primary source of variability for H3K4me2 coincides with the difference between UI and PI conditions (PC1 ~ 44%). Therefore, genome-wide epigenomic changes take place upon 1° infection. The second source of variability (PC2 ~ 27%) corresponds with the distance between the two biological replicates of the 1° infection time point, possibly underlying the heterogenous response of the epigenome after the 1° infection. Taken together, these data show that upon infection, a large number of peaks differ in comparison to UI, and this difference continuously increases up to 7 days (PI). Representative examples of peaks undergoing an increase in H3K4me2 are shown in Fig. 6E.

We further clustered the enrichment of H3K4me2 for the most dynamic peaks, i.e., those showing the biggest changes in methylation (Fig. 6C), and separated them into two global profiles, those that gain and those that lose signal intensity compared to UI. These profiles reveal that the gain or loss observed following infection is maintained at 7 days (PI). Interestingly, we compared this H3K4me2 methylation profile to that of ChIP-seq we performed on H3K4me3 and observed a very different peak pattern. Indeed, tri-methylation peaks were detected rapidly after infection (1°), but disappeared, leaving no differential peaks at 7 days (PI) compared to UI. These results further support that pneumococcal infection specifically induces di-methylation, a modification that is maintained genome-wide for at least 7 days (PI).

We proceeded to map the localization of the most significant H3K4me2 peaks, which we name "differentially methylated" (DM), that were detected post infection compared to those detected in uninfected conditions (Fig. 6D). Under these stringent conditions, we mapped 375 peaks, 173 from UI and 202 for PI conditions. Interestingly, although H3K4me2 in uninfected cells is lowly abundant in Inter-Genic regions (InterG), and most abundant in Intra-Genic (IntraG) and at transcription start sites (TSS), distribution is altered upon infection. In PI conditions, most peaks are present at inter and intragenic regions, and not at transcription start sites. By incorporating ENCODE data into our data on DM peaks, we performed unbiased clustering according to chromatin state, profile, and localization (Fig. S6A). Strikingly, peaks from UI conditions clustered together according to their localization to TSS and regions known to accumulate H3K4me3. These peaks are also shared with our ChIP-seq data on H3K4me3. In contrast, PI peaks are unique and cluster in regions known to have marks of enhancer regions (H3K27ac and H3K4me1). Therefore, altogether, our ChIP-seq data show that 7 days (PI), infection has remodeled the host methylome as cells have differential peaks to UI conditions, and this process occurs mainly in enhancer regions of the genome.

### Identification of the regulatory modes underlying primary and secondary infections

Even though our ChIP-seq data identified a significant number of di-methylated peaks 7 days post infection, at this same time point, there are no gene expression changes between UI and PI conditions. However, upon 2° infection, gene expression is altered in a significant manner compared to 1° infection, raising the question of a possible link between H3K4me2 peaks and transcriptional changes. To address this point, we integrated our transcriptome data with our ChIP-seq data in order to find links between the two analyses (Fig. S6B). A direct intersection of the differentially methylated peaks (DMs) with the differentially regulated genes (DE) only yielded 29 direct associations (dark line continuous in Fig. S6B). Amongst these, only 9 were linked with a gain in methylation, which is what we are particularly interested in. Therefore, we decided these few genes were not sufficient to

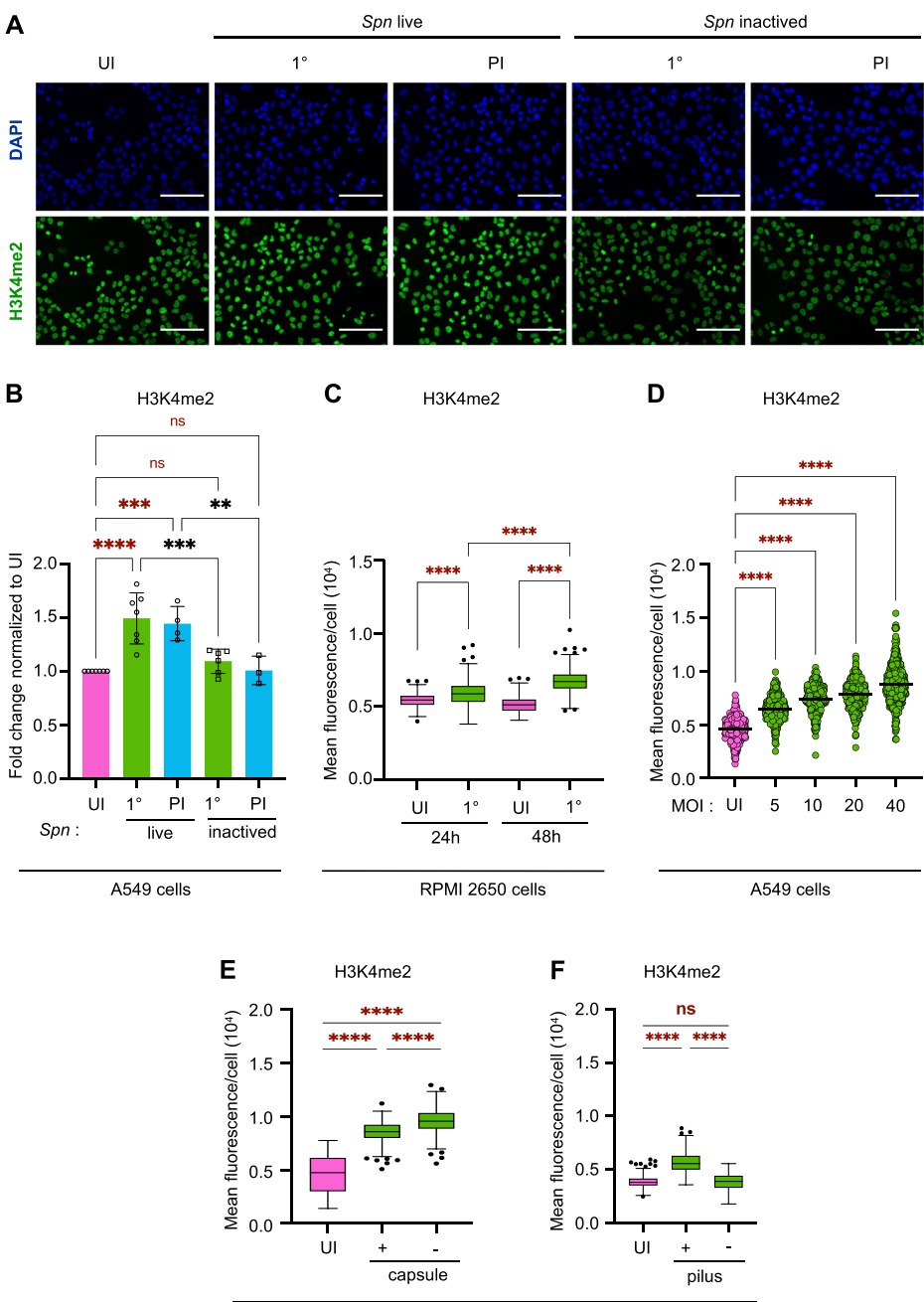

**Fig. 4 | The increase in H3K4me2 levels is actively induced by live bacteria.**
**A** Representative images of immunofluorescence detection of nuclear H3K4me2 in A549 UI or infected with *Spn* live and *Spn* inactived (1° and PI, MOI 20) at time γ. Cells stained for H3K4me2 (GFP; green) and DAPI (blue). Microscopy images were taken at ×20 magnification, and the scale bar represents 100µm. **B** Quantification of H3K4me2 normalized to the segmented nuclei using DAPI signal. Data points expressed as fold change of mean fluorescence intensity of infected cells 1° and PI to UI (MOI 20) with *sp* live and *sp* inactived at time g. Graphs display quantification from $n = 3$ to 7 biological replicates with the mean values ±SD for each condition. Statistical significance was determined by one-way ANOVA with Fisher's LSD test (ns = not significant, **$p = 0.0012$, ***$p = 0.0001$, ****$p < 0.0001$). **C** Quantification of H3K4me2 intensity normalized to the segmented nuclei using DAPI signal at time β and γ 1° (MOI 10) and UI. Box and whiskers plot with line denoting the median value from $n = 754$ to 1075 nucleus by conditions in the same experiment. Statistical

significance was determined by one-way ANOVA comparing means with Tukey's multiple comparison test (****$p < 0.0001$). **D** Quantification of H3K4me2 according to Multiplicities of infection (MOI) normalized to the segmented nuclei using DAPI signal at time γ. Histogram shows the mean fluorescence intensity ±SEM from $n = 435$ to 480 nuclei by conditions in the same experiment. Statistical significance was determined by one-way ANOVA with Fisher's LSD test (****$p < 0.0001$). **E, F** Quantification of H3K4me2 intensity after infection with **E** *Spn* mutant without capsule (−) or **F** *Spn* mutant without pilus (−), compared to wildtype *Spn* (+). H3K4me2 intensity is normalized to the segmented nuclei using DAPI signal at time γ post 1° (MOI 20) and UI. Graphs display quantification $n = 300$ nuclei/condition/exp. Box and whiskers plot with line denoting the median value. Statistical significance was determined by one-way ANOVA comparing means with Tukey's multiple comparison test (****$p < 0.0001$). Source data are provided as a Source Data file for **A**–**F**.

further our analysis, and we extended our analysis to include regions of methylation and genes beyond the strict statistical significance of our epigenome and transcriptome studies (black lines dotted in Fig. S6B). Using this method, we predicted 3845 regulatory links between H3K4me2 regions and genes. Together with the above 29, we analyses 3874 associations from a global perspective, using a multiple factor analysis (MFA)[20]. This factorial method is aimed at uncovering the main sources of variability that characterize a dataset among multiple sets of quantitative and qualitative variables. For the analysis, we used both transcriptome and methylome, more precisely: the complete transcriptome dataset of the UI cells, post 1° infection, PI, and post 2° infection (TOME); the H3K4me2 log fold change of PI vs UI and the average H3K4me2 level of the 1° infection time course (METH). We added the epigenome profile of the UI cells (EPIG) described by six key histone marks symptomatic of the main active and repressive chromatin states (CS). We retrieved ten ENCODE samples H3K4me3, H3K4me2, H3K4me1, H3K27ac, H3K79me2 and H3K27me3 from the ENCODE database (https://doi.org/10.1038/s41586-020-2493-4) for A549 cells. Finally, we considered two additional variables that describe the regulatory links between genes and H3K4me2 peaks, as estimated by the T-Gene tool of the MEME suite (https://doi.org/10.1093/bioinformatics/btaa227): the distance in bp separating each gene-peak pair (DIST) and the correlation between chromatin and gene expression changes in an independent tissue panel (CORR) (Fig. S7A).

Factors resulting from the MFA summarize the structure of the joint groups of variables, as shown by the cumulative percentage of variance associated with the first 5 factors/dimension (Dim) and the correlation coefficients of groups of variables with each factor (Fig. S7B). Such factors unveil correlations among the different groups, one of which is very interesting for our study, namely the correlation between DMRs and DEGs in Dim1 and that between TOME and METH in Dim3. (Fig. S7B and Fig. 7A). Furthermore, in the methylome analysis, an increase in dynamic H3K4me2 peaks at 7 days (PI) was demonstrated. This correlation suggests a strong link between an increase in gene expression levels and the increase in H3K4 di-methylation at 7 days (PI). Additionally, Dim2 unveils a correlation between the EPIG, CS, and DMR, further supporting the H3K4me2 epigenome profile at enhancer region (Fig. S6A).

Using the top factors obtained by the MFA, we clustered the regulatory links and characterize them using the variables enriched among dynamically marked H3K4me2 region, differentially expressed genes, and the functional categories specifically enriched. This analysis allows a subclassification of the data into smaller clusters unified by common features (Fig. 7B, and S7C). We focused our attention on clusters 1, 2, 4, 5, and 9 that contain most of the peaks gaining H3K4me2 methylation. Representative examples of H3K4me2 regions from each cluster are shown in Fig. 6E. MFA cluster 9 is the only cluster including H3K4me2 peaks that are differentially methylated in a significant manner between UI and PI conditions, while all other clusters include all H3K4me2 peaks. Such an analysis reveals that most H3K4me2 peaks gained in active enhancers are associated with downregulated genes, either specific to 1° infection (MFA clusters 1 and 2), or in a nonspecific manner (MFA cluster 9). At the 1° infection, we also find peaks associated with transcriptionally active promoters, but at downregulated genes, further supporting a role of H3K4me2 in transcriptional repression (MFA cluster 4). Thus, H3K4me2 seems to play a role in gene downregulation, and GSEA analysis of these, attributes their function to chromatin organization and cell response to external stimuli. Furthermore, it is interesting to note that the functional analysis of MFA clusters 1 and 2 (only 1° infection) reveals a role in vesicle-mediated transport and supports our LAMP1 data showing more LAMP1-positive lysosomal compartments detected in the 1° infection. Peaks associated with active enhancers exclusively mark genes specific to 2° infection regardless of up or downregulation (MFA

cluster 5). Interestingly, functional analysis of this cluster displays a role in metabolism and comfort our data in Fig. 2C exhibiting distinct metabolism dynamics upon 1° and 2° infections. Therefore, although straightforward global links between transcriptome and epigenome yielded a very small number of genes, MFA generated meaningful associations by clustering our data into smaller clusters. These reveal that infection-induced H3K4me2 probably has different roles depending on the genomic localization and the transcriptional context in which it is placed.

We reasoned that each MFA cluster could be regulated by different transcription factors. Therefore, we performed a transcription factor motif enrichment analysis on the H3K4me2 peaks of clusters gaining marking upon infection, i.e., MFA clusters 1, 2, 4, 5, and 9 (Fig. 7C). Interestingly, we obtained the enrichment of TFs known to be involved in immune regulation, such as SMAD3, and IRF and STAT family members, all shown to have a role during infection by viral and bacterial pathogens. Other TFs families, such as Fox, Hox, Zinc finger, OTX, and CREB have more extended cellular functions. Remarkably, this analysis reveals significant motif enrichment for the Activator Protein-1 (AP-1) family members. This ubiquitous dimeric protein complex composed of different Jun and MAF subfamilies, has been shown to play a role mediating inflammation and in transcriptional memory responses[21–23]. Therefore, S. pneumoniae infection modifies host chromatin in an impactful way by impacting major transcription factor regulatory processes.

## Discussion

In our study, we demonstrate that infection by S. pneumoniae results in a long-lasting modification of epithelial cells characterized by changes in transcription, phenotype, and a specific histone modification, H3K4me2. This modification is detectable shortly after infection, specific to di-methylation of lysine 4, and persists for at least 9 days after bacteria are cleared with antibiotics. Genome-wide analysis of H3K4me2 peaks shows that they are mainly localized to enhancer regions in which pioneering transcription factors are predicted to bind, and when correlated with transcriptional regulation of nearest genes, these are mostly repressed. Interestingly, the observed epigenetic modifications arise upon the interaction of epithelial cells with live bacteria, indicating an active pathogenic mechanism, resulting in a higher bacterial load upon secondary infection.

Bacterial presence or production of secondary metabolites are types of stimuli sensed and to which eucaryotic cells respond through the integration of signal transduction resulting in chromatin rearrangements and transcriptional modulation. However, certain bacteria impose their own infection-favoring histone modifications to reprogram host transcription[5,6,24]. To target chromatin, bacteria have been shown to either hijack and repurpose host enzymes or inject factors termed nucleomodulins, which will directly modify histones[3]. We had previously shown that pneumococcus dephosphorylates histone H3S10 by relocalizing the host PP1 phosphatase to promote infection, and that the pneumolysin toxin PLY and pyruvate oxidase SpxB were the main bacterial factors mediating this modification[13]. However, we had not assessed the lasting potential of this mark. In our current work, we show that H3S10 dephosphorylation is a transient modification that rapidly returns to pre-infection levels shortly after clearing of bacteria and is not a precursor mark for H3K4me2. In contrast, H3K4me2 is acquired during infection and persists in multiple cell passages post-antibiotic treatment. This is the first characterization of a mark that is maintained and affects epithelial cell physiology over time. Interestingly, we show that pili and bacterial adhesion to cells is necessary to induce H3K4me2, whereas capsule is not. Furthermore, we show that di-methylation is an active mechanism that only occurs if bacteria are alive. In combination with our finding that the impact on a secondary infection is to increase bacterial numbers

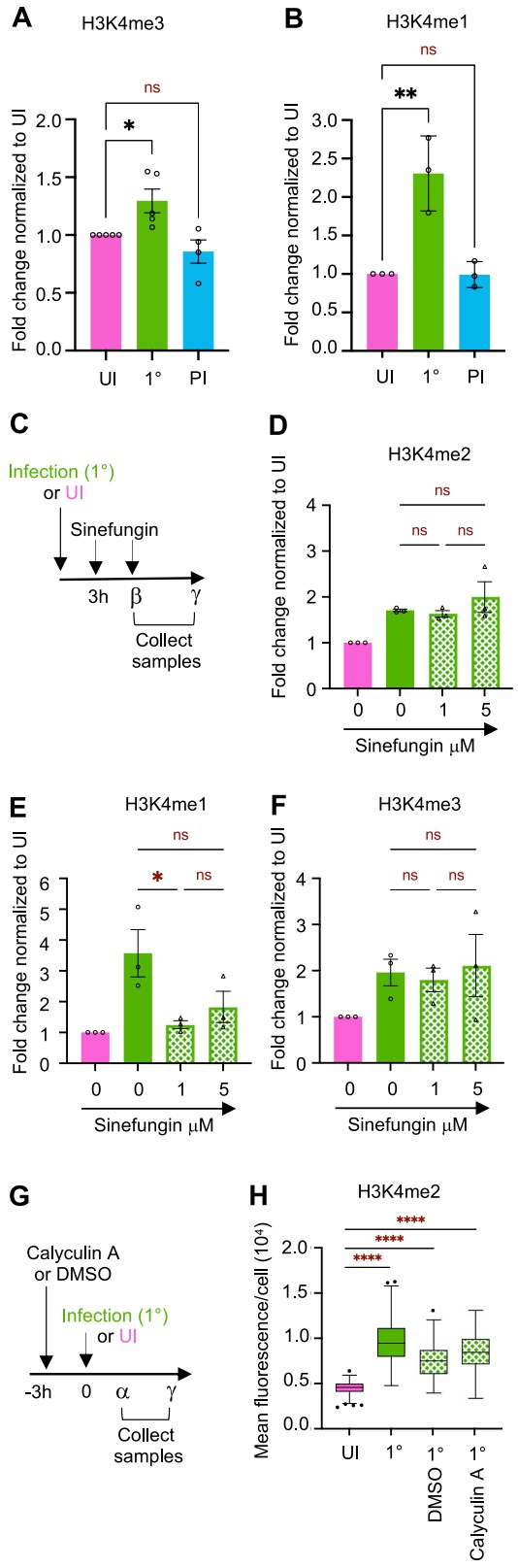

**Fig. 5 | H3K4me2 is a specific persistent mark induced by infection.**
**A** Quantification of H3K4me3 normalized to the segmented nuclei using DAPI signal. Data points are expressed as fold change of mean fluorescence intensity of infected A549 cells (MOI 20) at time β post 1° and PI to UI. Graphs display mean ±SEM from $n = 4$ to 5 biological replicates. Statistical significance was determined by one-way ANOVA with Fisher's LSD test (ns = not significant, *$p = 0.0208$).
**B** Quantification of H3K4me1 normalized to the segmented nuclei using DAPI signal. Data points are expressed as fold change of mean fluorescence intensity of infected A549 cells (MOI 20) at time γ post 1° and PI to UI. Graphs display mean ±SD from $n = 3$ biological replicates. Statistical significance was determined by one-way ANOVA comparing means with Tukey's multiple comparison test (ns = not significant, **$p = 0.0041$). **C** Experimental set-up with methyltransferase global inhibitor Sinefungin describes in Methods section (β = 24 h, γ = 48 h).
**D–F** Quantification of H3K4me2 at time γ (**D**), H3K4me1 at time γ (**E**), H3K4me3 at time β **F** normalized to the segmented nuclei using DAPI signal. Data points are expressed as fold change of mean fluorescence intensity of infected cells (MOI 20) at 1° to UI. Graphs display the mean values ±SEM from $n = 3$ biological replicates. Statistical significance was determined by one-way ANOVA comparing means with Tukey's multiple comparison test (ns = not significant, *$p = 0.0323$). **G** Experimental set-up with Calyculin A, described in Methods section (α = 3 h, γ = 48 h).
**H** Quantification of H3K4me2 at time γ, intensity normalized to the segmented nuclei using DAPI signal at time γ post 1° (MOI 20) and UI. Box and whiskers plot with a line denoting the median value from $n = 3$ biological replicates. Statistical significance was determined by one-way ANOVA comparing means with Tukey's multiple comparison test (****$p < 0.0001$). Source data are provided as a Source Data file for **A**–**H**.

KDM6B, to drive a homeostatic host response[12]. Disruption of this process leads to host cell damage in vitro and in vivo dissemination of the asymptomatic strain to the circulation. Although the effect of KDM6B is not on H3K4, as it is not a direct substrate, it is interesting to think that similarly to the histone modification induced by KDM6B, di-methylation could be important to better colonize respiratory epithelial cells. Whether H3K4me2 methylation is a virulence mechanism driving infection, or a means to enhance asymptomatic colonization remains to be determined.

The respiratory epithelium is composed of three major cell types: ciliated, secretory, and basal cells. These play critical roles in providing a barrier and defense system against inhaled pathogens[25]. The cells in contact with invading bacteria are thought to be terminally differentiated and constantly renewed. In our study, we show that H3K4me2 is induced in terminally differentiated type II pneumocytes (A549 cells), but also in undifferentiated nasal cells (RPMI cells) and in an in vivo model of infection. These results indicate that pneumococcus-induced epigenetic modifications occur in precursor cells and, therefore, have the potential to be transmitted to cells of the epithelium over multiple generations. Furthermore, we show that H3K4me2 is maintained through cell division. We hypothesize that H3K4me2 occurs initially in cells in contact with pneumococcus, and this mark is transmitted through cell division. Therefore 7 days later, cells display H3K4me2 even if they were not initially in contact with bacteria. Epigenetic transmission is more likely than the possibility that continuous cellular signals perpetuate the mark over time as we no longer detect any live bacteria after 24 h, and no more bacterial products after 3 days. Furthermore, cell divisions over time ensure that after 7 days cells are in majority renewed. Therefore, we show that *S. pneumoniae* epigenetically modifies epithelial cells during infection, and the epigenomic mark is transmitted through cell division to cells that were not in contact with bacteria, illustrating the potential to modify large numbers of cells.

H3K4 has the potential to be mono-, di- or tri-methylated and the number of methyl groups could define different chromatin states. H3K4me1 is a well-established feature of enhancers, often associated with active enhancers, and H3K4me3 marks the promoters of active genes[26]. H3K4me2 is less well defined, and often characterized along with H3K4me3 as functional studies are done by inactivating a

approximately 10-fold, our results indicate that this process is a mechanism driven by pneumococcus for its own benefit, and not a host stress response.

Bacteria-induced histone modifications have often been associated with virulence mechanisms as mostly pathogenic bacteria have been studied. However, we have recently shown that an asymptomatic pneumococcal colonizing strain activates a histone demethylase,

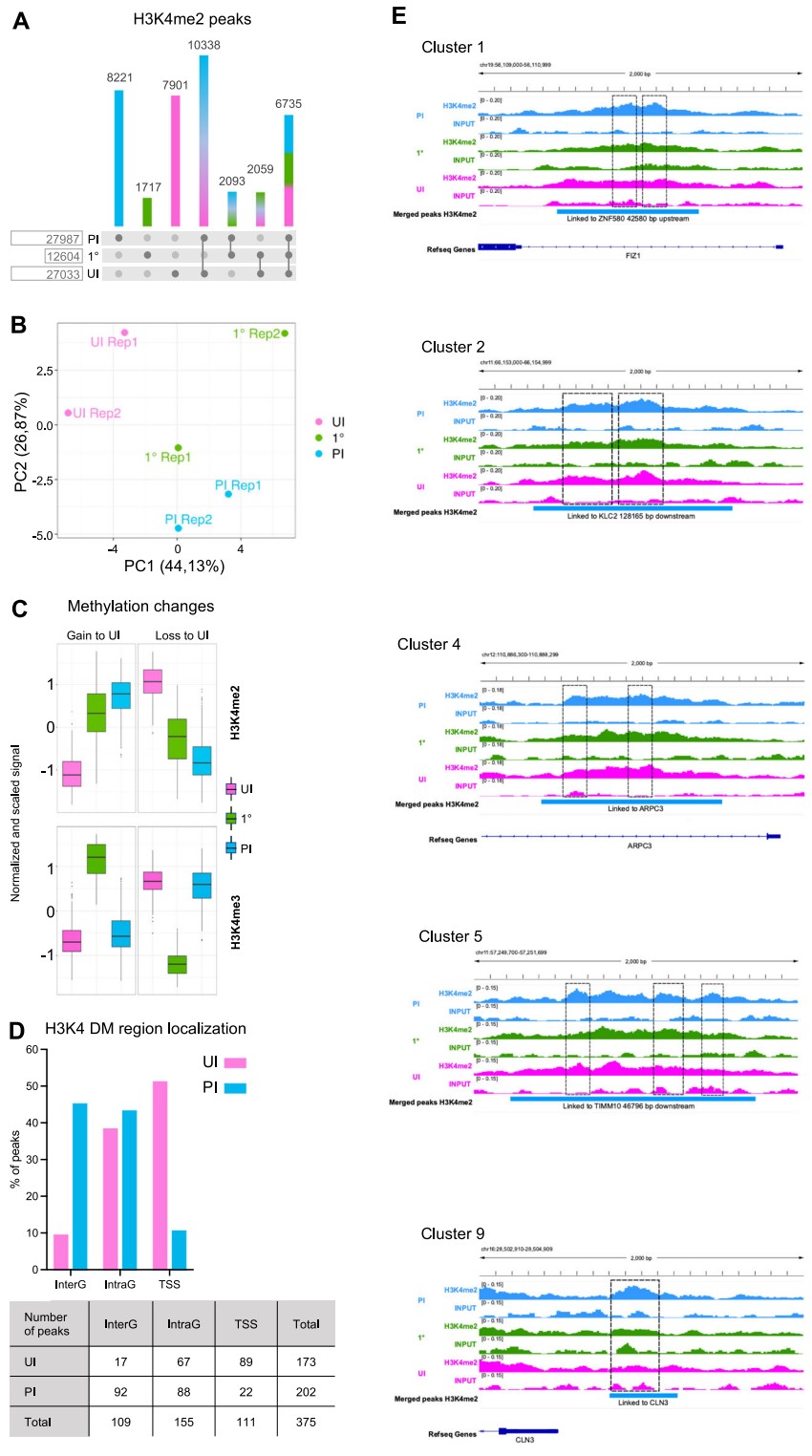

**Fig. 6 | Methylome dynamics during primary infection. A** Number of reproducible peaks between replicates per time point are compared among the three-time points by simple overlap of at least 50% of the peak length. **B** Principal component analysis of the H3K4me2 peaks by sample. Principal components are calculated on peak enrichment, i.e., read counts, after removal of the replicate batch effect. **C** Methylation changes for H3K4me2 and H3K4me3. Count distribution is plotted per time point for the most dynamic regions (DMRs), i.e., those showing the biggest changes in methylation. Read counts per peak are normalized by sequencing depth, peak length, and the replicate batch effect is corrected; they are subsequently centered and scaled by their standard deviation. Peaks are clustered and two global profiles are shown: gain and loss of methylation with respect to UI **D** H3K4me2 DMRs localization. Percentage and number of UI peaks (loss of

methylation) and PI peaks (gain of methylation) according to the localization: Transcription start site (TSS) for peaks overlapping the 2 Kb interval centered around the transcription start site; Intra Genic (IntraG) for peaks located within the gene annotations and outside the TSS interval; Inter Genic (InterG) for all other peaks. **E** Genome browser snapshots of representative regions (windows of 2 Kbp), gaining H3K4me2 over the time course. Tracks show the normalized coverage, pooled over the two replicates for H3K4me2 ChIP and INPUT samples. Genes predicted to be linked to the peak (see methods) are represented within the window when possible, otherwise their name and distance to the peak are indicated. Clusters are defined with the Multiple Factor Analysis (see Fig. 7). Source Data **A**–**E** were deposited into the Gene Expression Omnibus (GEO) repository of the National Center for Biotechnology Information under accession number GSE230142.

methyltransferase, most often shared between di- and tri-methylation[27,28]. In yeast, H3K4me2 has no clear association with transcriptional status, in animals H3K4me2 was suggested to be involved in transcriptional activation, and in plants the association is with repression[29]. As genomic localization and association with other histone modifications and chromatin features are important for defining the role of a particular mark, it may be that a generalized role for H3K4me2 cannot be made. Alternatively, heterogeneity in the cell population could make bulk analysis difficult to extract overarching features. This is what we find in our analysis, where the direct genome-wide association of H3K4me2 peaks with transcriptional activity was not possible on the global level. Instead, unbiased MFA allowed us to divide our findings into smaller clusters where a combination of chromatin states, genomic localization, and transcriptional regulation allowed us to draw some common features. This type of analysis has not previously been used to link transcriptome and epigenome data, but could be a good solution as the integration of ChIP-seq and RNA-seq data remains difficult and is slowly being addressed[30]. Intriguingly, if H3K4me2 was a "true" epigenetic mark, its function would be in maintaining chromatin states and not directly on transcriptional regulation.

Epigenetic transcriptional memory has been documented in multiple organisms and is thought to play a role in adaptation to changes in environmental conditions. In plants upon drought stress, a subset of genes display an increased transcription rate upon re-exposure to dehydration, along with a maintenance of high levels of H3K4me3 and phosphorylation of polymerase II at serine 5[31]. Similarly, hyperosmotic priming of *Arabidopsis* seedlings led to enhance drought tolerance in plants correlated with alterations in H3K27me3 profiles[32]. In yeast, localization within the nucleus at the periphery, as well as association with the histone variant H2A.Z, allows for robust reactivation[33]. In mammalian cells, one report shows that in HeLa cells, certain Interferon-γ regulated genes accumulate persistently high levels of H3K4me2, correlating with facilitated promoter accessibility for multiple cell cycles[34]. In all these studies, the histone mark is shown to be transmitted through the cell cycle and maintained through multiple cell divisions. Our results fit perfectly with already published data on epigenetic transcriptional memory. Infection specifically induces H3K4me2, which we identified by screening 21 different histone H3 marks and is maintained through multiple cell divisions affecting transcription upon secondary infection.

Innate immune memory (also known as trained immunity) is the process by which innate immune cells such as macrophages, monocytes, and natural killer cells, mount a faster and increased response to homologous or heterologous secondary challenge and display associated histone modifications[15]. Our data do not fit innate immune memory patterns as antibacterial responses are not elevated compared to the primary infection, live bacteria are essential, and a greater number of bacteria are recovered upon secondary infection. Thus, our data indicate that bacteria have their own mechanism of inducing transcriptional memory, and therefore pathogen manipulation of innate immune memory needs to be further studied to understand the impact on infectious processes.

By analysis of ENCODE data, one report has found that 90% transcription factor binding sites overlap specifically with H3K4me2 enriched regions, suggesting a tight regulation of H3K4me2 by transcription factor binding[35]. In our study, we performed a transcription factor motif search and found enrichment of certain binding sites in a MFA cluster-dependent manner. This result is interesting as it suggests there are specific genomic regions targeted by H3K4me2, which by Reactome analysis have biological significance for infection. Whether these transcription factors are necessary to initiate memory or maintain it remains to be determined. In yeast, transcriptional memory of the INO1 gene has been shown to depend on H3K4me2 and initiated by a particular transcription factor Sfl1, which binds to a subset of genes

upon repression of their transcription[36,37]. Interestingly, our MFA analysis links most gain of methylation peaks with genes down-regulated upon primary infection. This could suggest that the transcription factors that we identify could act similarly to Sfl1 in binding to repressed genes thereby initiating memory at these loci. Alternatively, a subset of the identified transcription factors could be maintained after bacterial clearance, thereby maintaining an open chromatin state and transcriptional memory. This mechanism would be similar to what has been reported in a skin inflammation model, where a combination of the transcription factors STAT3 and FOS are transiently induced, but only JUN remains bound to memory domains[21].

Although some molecular aspects remain to be determined, we provide a demonstration that bacteria modify host chromatin in a lasting manner, and for their benefit. Characterizing this process could have important consequences for our understanding of bacterial infection and/or colonization, particularly important for *S. pneumoniae*. Importantly, as epigenetic mechanisms are reversible, targeting the processes exploited by bacteria could be a prolific and novel strategy to develop host-directed therapies.

## Methods

### Ethics statement
All protocols for animal experiments were reviewed and approved by the CETEA (Comité d'Ethique pour l'Expérimentation Animale−Ethics Committee for Animal Experimentation) of the Institut Pasteur under approval number dap170005 and were performed in accordance with national laws and institutional guidelines for animal care and use.

### Cell culture condition
The human alveolar epithelial cell line A549 (ATCC CCL-185) was cultured in F-12K culture medium (Gibco™ 21127030) supplemented with 10% fetal calf serum (FCS) and 1% L-glutamine. The human nasal septum epithelial cell line RPMI 2650 (ATCC CCL-30) cells were cultured in Advanced MEM culture medium (Gibco™ 12492013) supplemented with 2,5% FCS and 4 mM L-glutamine and incubated at 37 °C in a humidified atmosphere with 5% $CO_2$. For assays, $1 \times 10^5$ A549 cells/mL or $2 \times 10^5$ RMI 2650 cells/mL were seeded in 6-well plates 2 days before infection. A549 cells were serum-starved (0.25% FCS) for 24 h before use in experiments.

### *Streptococcus pneumoniae* culture and cell infection
*S. pneumoniae* strains used in this study are all TIGR4 serotype 4, from gift of Thomas Kohler (CI50) and from gift of Jeffrey Weiser (CI187), and mutant strains derived CI187, unencapsulated (CI188), pilus-1 knock out (CI189)[38]. Experimental starter stocks were prepared on 5% Columbia blood agar plates (Biomerieux 43041) from frozen permanent stocks. Bacteria are grown Todd−Hewitt (BD) broth supplemented with 50 mM HEPES (TH + H) at 37 °C with 5% CO2 to mid-exponential phase 0.6 $OD_{600}$. Aliquots were made in TH + H media supplemented with Luria−Bertani (BD) and 15% glycerol final concentration and frozen at −80 °C. All experiments were performed with frozen experimental starters of *S. pneumoniae* (*Sp*) less than 1 month old. For experiments, starters were grown to mid-exponential phase (0.6 $OD_{600}$) in TH + H broth at 37 °C with 5% CO2, washed twice in PBS and concentrated in 1 ml PBS. For quantification of LAMP1 and colocalization with bacteria (Fig. S2C), bacteria are staining with 100 µg/mL Wheat Germ Agglutinin conjugates Alexa Fluor 488 (Invitrogen™ W11261) in buffer (2.5% BSA, 0.15 M NaCl, PBS) 30 minutes at 37 °C, 5% CO2. Then two washes with staining buffer and PBS. Bacteria were diluted in serum-low cell culture medium (0,25% FCS) to get a multiplicity of infection of 20:1 (MOI 20), 10:1 (MOI 10) or 35:1 (MOI 35). After 1 h or 3 h of *Sp* infection, cells were washed twice with PBS to remove the unattached bacteria, and then either collected for analysis, or cultured in medium with penicillin-streptomycin (10 µg/mL) and gentamicin (200 µg/mL for A549 cells and 25 µg/mL for RPMI 2650 cells) for later time points.

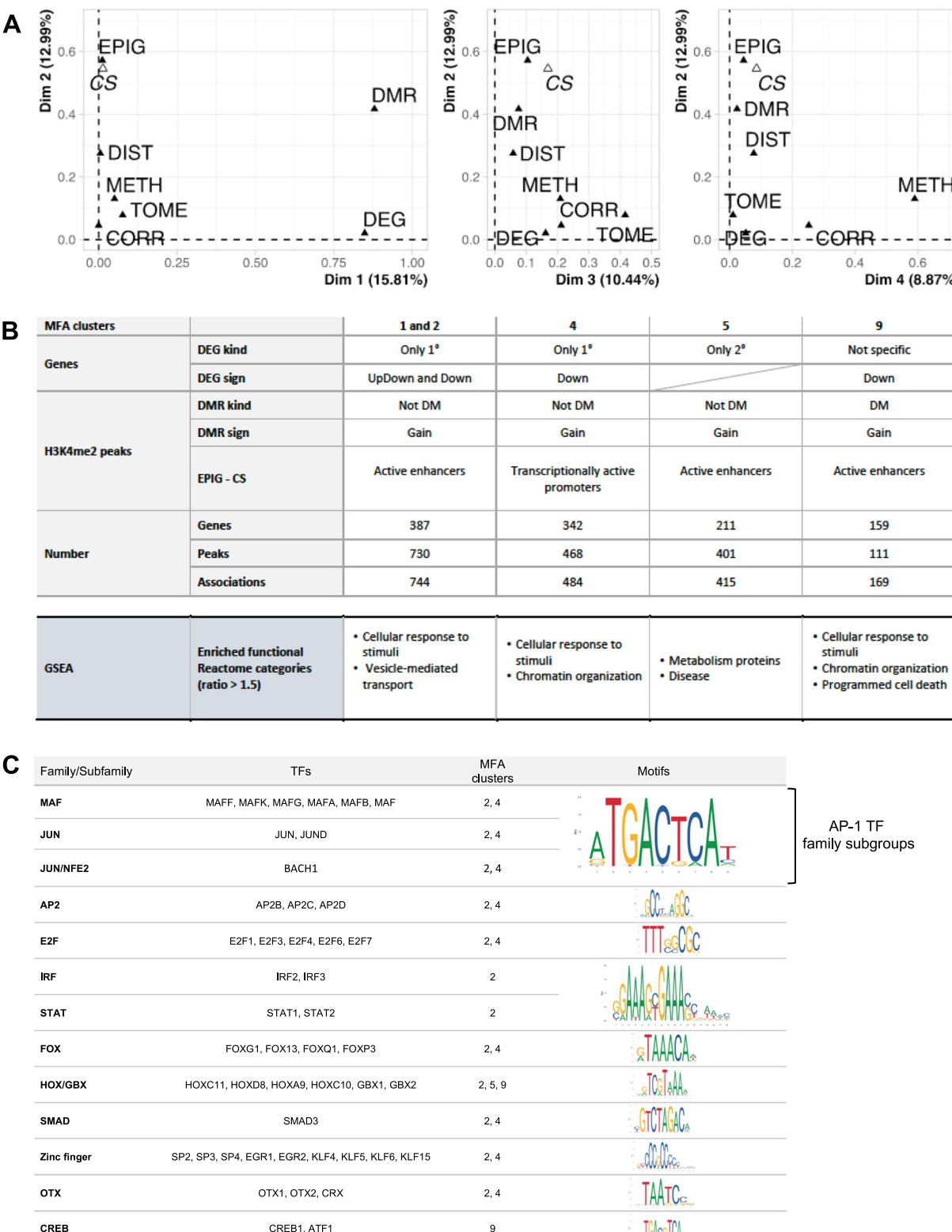

**Fig. 7 | Identification of the regulatory modes underlying the primary and secondary infections. A** Contribution of groups of variables to the definition of factors/dimensions. Groups of variables constitute the multiple factor analysis (MFA) input matrix and describe the genes (TOME = Transcriptome, DEG = Differentially expressed gene), the H3K4me2 peaks (Methylome = METH, DMR = Differentially methylated region, EPIG = Chromatin profile, CS = Chromatin state) and their association (DIST = distance, CORR = correlation). **B** Description of the four regulatory modes associated with an increase of H3K4me2. Clusters of gene-peak associations are characterized in terms of the categorical variables used in the MFA (DMR, DEG), the chromatin state of the peaks (EPIG, CS). Representation of enriched functional Reactome categories from GSEA for each MFA cluster: the observed number of genes belonging to a given Gene Set and the expected number per cluster (with ratio >1.5). **C** Transcription factors (TFs) associated with the regulatory modes. Table lists the TFs with binding motifs enriched in H3K4me2 peaks from the selected MFA clusters. Source Data A–C were deposited into the Gene Expression Omnibus (GEO) repository of the National Center for Biotechnology Information under accession number GSE230142.

To determine the amount of *Sp* that have colonized cells, PBS washed cells were lysed using sterile ddH2O. Lysates and serial dilutions were plated on 5% Columbia blood agar plates overnight at 37 °C with 5% CO2, and the colony-forming units (CFUs) were counted.

For inactivated bacteria with paraformaldehyde (PFA), the pellet of bacteria was incubated in 4% PFA for 20 min at room temperature, washed twice in PBS.

2 days after infection, A549 cells were recovered with PBS supplemented with 10 mM EDTA, then $1 \times 10^6$ cells were seeded in flask (Corning™ 430641U) with medium including penicillin-streptomycin (10 µg/ml) and gentamicin (200 µg/ml) (Fig. 1A).

For 2° infection or cells maintained post 1° infection (PI) assays, $1 \times 10^5$ A549 cells/mL were seeded in 6-well plates 2 days before infection. Cells were serum-starved (0.25% FCS) for 24 h before use in experiments (Fig. 1A).

### Cell viability with calcein AM and DRAQ7™ by FACS

After 3 h and 48 h infection, the cells were recovered with PBS supplemented with 10 mM EDTA. Cells ($5.10^6$), then were washed in PBS and were resuspended in 1 mL PBS in BD Trucount™ tubes with beads (663028, batch 21280). Viability staining using 0.05 µM Calcein AM (L3224A) and 0.3 µM DRAQ7™ (51-9011172) for 15 minutes at room temperature in dark, then at 4 °C. Acquisition of the different samples was performed on CytoFLEX S (Beckman Coulter) flow-cytometer. The resulting bivariate frequency distribution shows the clear separation of the green fluorescent (FITC, 530 nm) for Calcein AM and (PE, 585 nm) for beads, the red fluorescent (APC, 655 nm) for DRAQ7™. The analysis was completed using FlowJo Software. The absolute number (cells/mL) of cells in a population was determined by dividing the number of positive cell events by the number of bead events, then multiplying by bead concentration divided by volume. Quadrant gates split the population into 4 subpopulations (Q1-4)): double positive population Calcein AM/DRAQ7™ (Q2), double negative population Calcein AM/DRAQ7™ (Q4), single positive cell Calcein AM (Q3) and single positive cell DRAQ7™ (Q1). The percentage of events failing within each Q is calculated with respect to the events that fall in the analyzed total population.

### Cell proliferation with CellTrace™ CFSE by FACS

24 hours after infection or no-infection, cells were incubated at 37 °C with 5% CO2 in serum-low cell culture medium with penicillin-streptomycin (10 µg/ml) and gentamicin (200 µg/ml) containing 2.5 mM CellTrace™ CFSE (Invitrogen™ C34554). After 6 h, 3 days and 5 days, the cells were recovered with PBS supplemented with 10 mM EDTA. Cells were washed in PBS in preparation for viability staining using fixable viability dye (eFluor780, ebioscience) for 5 minutes at 4 °C. Cell were washed in PBS and fixed using commercial fixation buffer (Biolegend). After washes, cells were resuspended in PBS, and the acquisition of the different samples was performed on MACSQuant (Miltenyi Biotec) flow-cytometer and analysis was completed using FlowJo Software.

### Cell viability alamarBlue assay

3 hours after infection, cells were incubated at 37 °C/% CO2 in serum-low cell culture medium with penicillin-streptomycin (10 µg/ml) and gentamicin (200 µg/ml) containing 10% alamarBlue reagent (Molecular Probes™ DAL1025) for 12 hours. 1 measure of fluorescence (Ex/Em 560/590 nm) was then read each hour using a Cytation 5 (BioTek) driven by the Gen5 3.11 Software. Fluorescence readings were blank corrected to wells containing only culture medium and results are expressed as fluorescence intensity.

### Inflammatory cytokines analysis by multiplex ELISA

The supernatants of cells at time α are collected and analyzed for the simultaneous detection of multiple human biomarkers biomarker with Luminex Discovery Assay (R&D systems bio-techne brand; LXSAHM-

29) using the Bio-Plex 200 systems (BioRad). The concentration is expressed in pg/mL for each sample.

### Immunofluorescence microscopy and Fiji or Imaris analysis

Cells were grown on acid-washed and ultraviolet-treated coverslips in 6-well plates. For quantification of LAMP1 and colocalization with Lysotracker®, after infection, cells were washed 2 times with PBS, 62.5 nM Lysotracker® Deep Red (Invitrogen™, L12492) in pre-warmed medium at 37 °C added for 1 h at 37 °C with 5% CO2. Then, the cells fixed in 2.5% PFA for 10 min at room temperature. For quantification of H3K4 methylation, after infection the cells fixed in 2.5% PFA for 10 min at room temperature. After 3 washes, cells were blocked in 5% BSA for 6 h at +4 °C, then permeabilized overnight in 5% BSA 0.5% Tween20 at +4 °C. Immunostaining was performed overnight at +4 °C with primary antibodies (H3K4me2 1:1000; Abcam ab32356) (H3K4me2 1:1000; EpiGentek A4032) (H3K4me1 1:5000; Abcam ab176877) (H3K4me3 1:200; Diagenode C15410003) (LAMP1 1:100; Abcam ab25630) diluted in 5% BSA 0.5% Tween20, then washed three times in 0.5% Tween20 and three times in PBS. Cells were incubated for 1 h at room temperature with Alexa Fluor 488 secondary antibody (1:1000; Invitrogen™ A11034 or A11029) and Alexa Fluor 555 secondary antibody (1:1000; Invitrogen™ A21424) in 5% BSA 0.5% Tween20, then washed three times in 0.5% Tween20 and three times in PBS. Cells were incubated for 5 min at room temperature with 300 nM 4,6-diamidino-2-phenylindole (DAPI) (Invitrogen™, D1306) diluted in PBS, then washed three times in 0.5% Tween20 and three times in PBS. The coverslips were rinsed briefly in distilled water and were mounted on slides using ProLong™ Gold antifade reagent (Invitrogen™, P36930). For quantification of LAMP1 and colocalization with Lysotracker®, images were acquired using confocal microscope LSM480 (Zeiss) driven by Zen Software and processing was done with Imaris x64 9.9.1 software. For quantification of H3K4 methylation, images were acquired using Cytation 5 (BioTek) driven by the Gen5 3.11 Software and processing was done with ImageJ2 2.14.0 software.

### Animal model

Experiments were conducted using C57BL6 females of 6 to 7 weeks of age. CD45.1 mice were maintained in-house at the Institut Pasteur animal facility.

**Mouse housing conditions.** Animal Facility: Animals are maintained in different animal facilities on the Institut Pasteur campus, each of which is licensed by the French Ministry of Agriculture (A 75-15-01-1 to -9, renewed in 2021). Each facility has central air conditioning equipment which maintains a constant temperature of 22 ± 2 °C (air is first cooled then warmed, and hygrometry is ~55% ±10%). Air is renewed at least 20 times per hour in animal rooms. Fluorescent light is provided with a 14:10 h light:dark cycle. In BIME-A3 facility, animals are kept either in individually ventilated cages (IVCs) or in isolators depending on the pahtogen class. All cages comply with European regulations in terms of floor surface per animal.

Care of animals: The bedding used for mice is spruce cracks. Food is purchased from manufacturer and provided as irradiated (10 kGy) pellets (10 mm in dia.). Water is deionized and autoclaved in bottles. Cages are cleaned in large cage washers. They are filled with spruce cracks ( + three dental rolls), stacked on dedicated transport rolls and autoclaved in double-door autoclaves. Animals are changed weekly. Mice are handled with forceps.

### Mouse infection

Mice were anaesthetized with a ketamine and xylazine cocktail (intramuscular injection) prior to infection. Mice were intranasally challenged with 20 µl of $6.10^5$ *Streptococcus pneumoniae* (a sub-lethal dose). Control mice received intranasal challenge of 20 µl of PBS. At 3 days post infection, mice were euthanized by CO2 asphyxiation at persistent

clinical score 3. Lungs were washed to remove RBC with 10 mL of PBS, followed addition of 3 mL of Dispase 50 U/mL (Corning™ 354235) that had been pre-heated to 37 °C. Lungs were collected and incubated in 1 mL Dispase 50 U/mL for 45 min in a water bath at 37 °C. Lungs were homogenized using gentleMACS™ C tubes (Miltenyi Biotec, 130-093-237) in cell dissociation buffer (5 mL DMEM (Gibco, 61965-026), 20 mM L-Glutamin (Gibco™, 25030123), 120 U/mL DNase I (Invitrogen™; 18047019)). The samples were run on a gentleMACS™ dissociator (Miltenyi Biotec, 130-096-427) programmed to m-lung-02-01. To purify epithelial cells, lungs homogenate was filtered through 100 µm stainer (Miltenyi Biotec, 130-098-463), and samples were centrifuge for 10 min at $200 \times g$. Cell pellets were suspended in 1 mL RBC lysis buffer 1X (Biolegend; 43031) and incubated for 3 min at room temperature to lyse remaining RBC. Cell pellets were washed with 20 mL PBS/FCS 1%, and samples were centrifuged for 10 min at $200 \times g$. Finally, the cell pellet was resuspended in 10 mL PBS/FCS 1%, filtered through 40 mm stainer (greiner Bio-one, 542040), and centrifuge 10 min at $200 \times g$. The cell pellet was resuspended in PBS/FCS 1% and the cells were stained with CD45.1 Monoclonal PE (0.1 µg; eBioscience™; 12045383) and CD326 (EpCam) APC (1:50; Miltenyi Biotec; 130-117-865) antibodies and incubated for 1 h at 4 °C. Samples were washed with 3 mL PBS/FCS 1% and centrifuged for 10 min at 200 g. Samples were then stained with Fixable Viability dye eFluor™ 780 1:1000 (eBioscience™, 65-0865-14) for 5 min at 4 °C, in the dark and washed with 3 mL PBS/FCS 1% and centrifuge for 10 min at $200 \times g$. Samples were Fixed in PFA 4% for 10 min at RT in the dark washed with 3 mL PBS/FCS 1%, and centrifuge for 10 min at $200 \times g$. Fixed cells were stained with H3K4me2 antibody (1:1000; Abcam ab32356), Isotype control Rabbit IgG monoclonal (Abcam; ab172730), Alexa Fluor 488 secondary antibody (1:1000; Invitrogen™ A11034) using the protocol previously described in the immunofluorescence section. Cells were analyzed with MACSQuant (Miltenyi Biotec) flow-cytometer, and analysis was completed using FlowJo Software.

## Histone H3 modification multiplex assay

For cells infected and uninfected, the histone extraction kit (OP-0006) and the EpiQuik™ Histone H3 modification Multiplex assay kit (P-3100) were purchased from Epigentek and protocol followed according to manufacturers' instructions. 100 ng of total histone proteins extracted from A549 cells uninfected and infected (MOI 20) was used.

## Immunoblotting and quantification

A549 cells uninfected or infected (MOI 20) were lysed using laemmli buffer (4% SDS, 20% glycerol, 200 mM DTT, 0.01% bromophenol blue, and 0.1 M, pH 6.8). Samples were sonicated for 4 sec, boiled for 10 min. Proteins were separated on 15% SDS-PAGE and were transferred to PVDF membrane under 2.5 A/25 V condition for 7 min using a semidry transfer system (Trans-Blot Turbo, BioRad). Transferred membranes were first blocked by TBS-Tween20 (0.1%) with 5% milk or 5% BSA, and then incubated overnight at 4 °C with primary antibodies for H3K4me2 (1:2000; Abcam ab32356), H3K4me3 (1:1000; Diagenode C15410003), H3S10ph clone MC463 (1:2000; Millipore 04-817), anti-γH2A.X (S139) (2OE3) (1:1000; CST 9718 S), anti-H2A.X (1:1000; CST 2595 S) or actin AC-15 monoclonal (1:10.000; Sigma A5441). Membranes were washed with TBS-Tween20 (0.1%) and incubated with secondary-HRP conjugated antibodies (1:10.000; Anti Rabbit IgG HRPO, Millipore BI2407; Anti Mouse IgG HRPO, Millipore BI2413C) for 1 h at room temperature. After washing with TBS-Tween20 (0.1%), the immunoreactive bands were visualized using Clarity ECL substrate (BioRad). Immunoblot signal was acquired on a ChemiDoc Imaging Systems (BioRad) and analyzed using Image Lab software (BioRad).

## Treatment of cells with inhibitors

For Calyculin A (Bio-Techne Tocris, 1336) inhibitor of PP2A and PP1 phosphatase, A549 cells were pretreated for 30 min before infection with 0.05 µM Calyculin A or equal amont of DMSO, then 2 washes with PBS. After infection, cells were assayed at time α (3 h) for H3S10ph and at time γ (48 h) for H3K4me2. For Sinefungin (Sigma-Aldrich, S8559) methyltransferase global inhibitor, A549 cells were treated with a first dose at 3 h (γ), followed by a second dose at 24 h (β) after infection, with 2 different concentrations of inhibitor: 1 or 5 µM. Cells were assayed at time γ (48 h) post infection for H3K4me2 and H3K4me1 or at time β (24 h) for H3K4me3.

## Statistical analysis

Statistical tests are reported in the figure legends. Appropriate parametric or nonparametric tests were used. Data plots and statistics were generated using Prism (version 9, GraphPad Software Inc.).

## RNA samples and sequencing

RNA was extracted from cells using RNeasy Mini Kit (Qiagen, 74104), and protocol followed according to manufacturers' instructions. Chloroform extraction/Na-Acetate 3 M (pH 5.5), isopropanol precipitated and washed 3 times in 70% ethanol prior to being suspended in molecular grade water. Extracted RNA quality was assessed and quantified with RNA 6000 Nano kit (Agilent, 5067-1511) using 2100 Bioanalyzer instrument (Agilent). The Affymetrix Human Gene array 2.1 microarrays were performed by Eurofins Genomics (France).

## Transcriptome analysis

Expression profile of the complete time course, i.e., UI, 1°, 2° infections and PI in triplicates, was measured using the Affymetrix Human Gene array 2.1 microarray.

Expression signal was normalized using the Robust Multichip Average (RMA) method, from the affy R package (doi: 10.1093/bioinformatics/btg405). Control and lowly express probes were filtered out and the replicate effect was removed using the ComBat approach implemented in the sva R package[39]. Replicate 3 from 1° infection was filtered out after principal component analysis. Differential expression analysis was performed using the limma method[40] and the following comparison evaluated: 1° infection vs UI cells, 2° infections vs PI and (2° vs PI) vs (1° vs UI), i.e. 2° infection vs 1° infection. Considering the four conditions: UI, 1°, PI and 2°, we tested the following contrasts:

- primary infection: 1° - UI
- secondary infection: 2° - PI
- between infections: ((2° - PI) - (1° - UI)).

Genes with an adjusted $p$ value < 0.05, for the first two contrast, or an adjusted $p$ value < 0.3, for the last one, were defined as differentially expressed (DEGs). The two first comparisons were used in the general characterization of the transcriptome, Fig. 1D. The last comparison, which includes genes whose change in the 2° infection in comparison to PI is significantly bigger/smaller than the change in the 1° infection in comparison to UI, was used in the MFA to classify gene changes as Only 1°, Only 2° or not specific (Fig. S7).

For the functional characterization, we performed a GSEA for each comparison, by ranking the genes with the t-statistic, which is the log2 fold change divided by its standard error. We used the Reactome database gene sets and the GSEA implementation in https://gitlab.pasteur.fr/hvaret/fgsa_scripts.

## ChIP sequencing assays

After infection or no-infection, the cells were washed with PBS in each well of 6-well plates (TPP 92006). Cells were cross-linked with 1% formaldehyde 8 min at room temperature, followed by quenching with 0,125 M glycine for 5 min. After two washes in PBS, cells were collected by scraping, then pelleted and lysed on ice for 5 min in 0.25% Triton X-100, 10 mM Tris-HCl (pH 8), 10 mM EDTA, 0.5 mM EGTA and proteases inhibitors. The soluble fraction was eliminated by centrifugation, and chromatin was extracted with 250 mM NaCl, 50 mM Tris-HCl (pH 8),

1 mM EDTA, 0.5 mM EGTA, and proteases inhibitors cocktail for 30 min on ice. Chromatin was resuspended in 1% SDS, 10 mM Tris-HCl (pH 8), 1 mM EDTA, 0.5 mM EGTA, and protease inhibitor cocktail, then fragmented by sonication (10 cycles of 30 sec 'on' and 30 sec 'off') using Bioruptor Pico (Diagenode). Sheared chromatin was cleared by centrifugation and tested for shearing efficiency analysis with High Sensitivity DNA kit (Agilent, 5067-4626) using 2100 Bioanalyzer instrument (Agilent). 2 µg of antibodies ChIP-grade (H3K4me2, Abcam ab32356; H3K4me3, Diagenode C15410003; H3, Abcam ab1791; Normal Rabbit IgG, Abcam ab37415) were used per ChIP and were bound to DiaMag protein G-coated magnetic beads (Diagenode, C03010021) overnight at 4 ˚C with gentle rotation. 6–7 µg/IP experimental chromatin was diluted 10 times in 0.6% Triton X-100, 0.06% sodium deoxycholate (NaDOC), 150 mM NaCl, 12 mM Tris-HCl, 1 mM EDTA, 0.5 mM EGTA and proteases inhibitors cocktail. 2% of ChIP sample volume was reserved to serve as input. Diluted chromatin was then added to antibody-bound DiaMag beads and incubated at 4 ˚C overnight with gentle rotation. ChIP samples were washed sequentially 5 min with buffer 1 (1% Triton X-100, 0.1% NaDOC, 150 mM NaCl, 10 mM Tris-HCl (pH 8)), buffer 2 (0.5% NP-40, 0.5% Triton X-100, 0.5 NaDOC, 150 mM NaCl, 10 mM Tris-HCl (pH 8)), buffer 3 (0.7% Triton X-100, 0.1% NaDOC, 250 mM NaCl, 10 mM Tris-HCl (pH 8)), buffer 4 (0.5% NP-40, 0.5% NaDOC, 250 mM LiCl, 20 mM Tris-HCl (pH 8), 1 mM EDTA), buffer 5 (0,1% NP-40, 150 mM NaCl, 20 mM Tris-HCl (pH 8), 1 mM EDTA) and buffer 6 (10 mM Tris-HCl (pH 8), 1 mM EDTA). ChIP and input samples were treated with IPure Kit (Diagenode, C03010015) following manufacturers' instructions. ChIP and input DNA were quantified with Qubit 4 fluorometer (ThermoFisher). Biomics Platform, C2RT, Institut Pasteur generated the libraries with 10 ng of each sample, and sequencing was performed with Illumina Hiseq2500.

## Methylome analysis

H3K4me3 and H3K4me2 were profiled using ChIP-seq for the UI cells, 1° infection, and PI in duplicates. Complete analysis from the raw sequencing data to the differential marking analysis was done following the ePeak approach[41]. Default parameters were used for the filtering, mapping of reads and for narrow peak calling. For H3K4me2, reproducible peaks were determined using the intersection approach, requiring at least 40% length overlap between replicates. For H3K4me3, reproducible peaks were determined using the IDR method.

Differential analysis was performed using the limma method[40] with a model considering the biological factor of interest, i.e., time (UI, 1° infection and PI) and the batch effect, i.e., replicate. Prior to the statistical test, the systematic differences between samples due to technical variation such as sequencing depth were normalized using the cyclic lowness method (i.e., locally fitting a smooth curve). The differential analysis was performed on the non-redundant set of reproducible peaks merged among all conditions.

Time course profiles were calculated using the read counts distribution over the dynamically marked regions (DMRs): adjusted $p$ value < 0.1 and log fold change > abs(0) for H3K4me3, adjusted $p$ value < 0.35 and log fold change > abs(0) for H3K4me2. Read counts over DMRs per time point and replicate were normalized to account for systematic technical variation (sequencing depth and replicate), differences in region length and they were subsequently scaled. Finally, DMRs were clustered using the Euclidean distance and the Ward agglomerating method to define the two main profiles: Gain and Loss of marking.

DMRs were classified according to their localization with respect to the transcriptional starting site (TSS) of the closest gene in: TSS if they overlap a window of 2 Kbp centered on the TSS; intragenic if they overlap the gene annotation but not the TSS window; intergenic if they overlap non-coding regions between gene annotations and don't overlap any TSS window.

Genome coverage was calculated using deepTools (doi:10.1093/nar/gkw257) for every ChIP and INPUT sample, both per replicate and

pooling replicates. Bins of 10 bp and counts per million reads normalization were used.

An epigenomic characterization of H3K4me2 DMRs was done for the UI condition using publicly available information for the cell line in the ENCODE portal (https://www.encodeproject.org/[42],).

We downloaded the call sets with the following identifiers: ENCFF628ANV, ENCFF510LTC, ENCFF556OVF, ENCFF250OYQ, ENCFF189JCB, ENCFF421ZNH (H3K4me3); ENCFF876DTJ, ENCFF975YWP, ENCFF419LFZ (H3K4me2); ENCFF137KNW, ENCFF103BLQ, ENCFF663ILW (H3K27ac); ENCFF165ZPD, ENCFF569ZJQ, ENCFF613NHX (H3K4me1); ENCFF892LYD, ENCFF154HXA, ENCFF931LYX (H3K79me2); ENCFF819QSK, ENCFF134YLO, ENCFF336AWS (H3K27me3). Coverage over peaks was normalized by length and scaled. Peaks with similar epigenomic profiles were clustered using the Euclidean distance and the Ward agglomerating method.

## Integrative analysis

Joint analysis of the complete dataset was performed by MFA[20].

Firstly, we predicted the regulatory links between DEGs and DMRs using T-gene[43] and selected the most significant links (distance <500 Kb and correlation $p$ value < 0.05). The joint dataset contains the 3874 links holding a differentially expressed gene (DEG) and/or a dynamically marked region (DMR). For each link, we defined five groups of continuous variables:

- Transcriptome (TOME) consisting of the gene expression over the four time points (UI, 1°, 2° infections and PI) in triplicates.
- H3K4me2 methylome (METH) measured by the average methylation over le three time points (UI, 1°, PI) and the log fold change between PI and UI cells.
- Epigenome of the UI condition (EPIG) including the coverage of the above-mentioned histone modifications over the H3K4me2 regions.
- Distance (DIST) and correlation (CORR) of the regulatory link between genes and regions.

and two groups of categorical variables:

- DEG kind and sing, describing whether genes are specific of the 1° or 2° infection, or not specific; and whether genes are going Up, Down, UpDown or DownUp along the time course.
- DMR kind and sing, indicating whether regions are DM or Not DM; and whether regions belong to the Gain or Loss of H3K4me2 profile along the 1° infection time course.

The chromatin state (CS) of each H3K4me2 region, calculated from the EPIG group as described in the previous section, was added as a supplementary variable to facilitate the interpretation of the MFA dimensions/factors. To control for the within-group variability, all continuous variables were normalized and scaled. MFA was performed, and a clustering of the 3874 was obtained using the top 10 dimensions/factors. Each cluster was then defined as a combination of the continuous and categorical variables more significantly associated to the genes and regions of the corresponding links (v.test statistic > 5).

## Transcription factor motif enrichment analysis

Transcription factor motif enrichment analysis was performed on the previously defined MFA clusters 1, 2, 4, 5, and 9, separating the peaks corresponding to a gain or loss of methylation. For each peak, 500 bp were extracted around the summit and then used as input for MEME-chip V5.1.1[44]. For the analysis, the HOCOMOCO v11 was used as the target database for the motifs, and the parameter ccut (maximum size of a sequence before it is cut down to a centered section) was set to 0. All the other parameters were the default parameters of the MEME-chip web version.

## Reporting summary

Further information on research design is available in the Nature Portfolio Reporting Summary linked to this article.

## Data availability

For transcriptome and methylome, data were deposited into the Gene Expression Omnibus (GEO) repository of the National Center for Biotechnology Information under accession number GSE230142. All these downstream bioinformatic and biostatistic analysis are available at: https://gitlab.pasteur.fr/cchica/epimem/. Source data are provided with this paper.

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

## Acknowledgements

We would like to thank Thomas Kohler, University of Greifswald, Germany, for his generous gift of *S. pneumoniae* strain and Jeffrey N. Weiser, NY University Langone Health, for his generous gifts of mutants and wild strain of *S.pneumoniae*. We would like to thank some members of Chromatin and Infection Laboratory, Tiphaine Marie-Noelle Camarasa for her help with the acquisition of cells for proliferation analysis, Julia Torné and Lucile Le Chevalier-Sontag for performing mice infections and lung collection. We would like to thank Pierre-Henri Commere of the Cytometry UtechS of C2RT, Institut Pasteur for his help, his precious advice and the acquisition of cells for viability analysis. We thank the staff of UTechS Photonic Bioimaging of C2RT, Institut Pasteur, for help, the training and advice for acquiring images with the confocal microscope LSM780. We thank Juliana Pipoli Da Fonseca of the Biomics Core Facility of C2RT, Institut Pasteur for processing the libraries and the sequencing of ChIP-seq. We would like to thank Esma Karkeni of the Biomarker UtechS of C2RT, Institut Pasteur, for processing the Multiplex Inflammatory cytokines protein quantification with Bioplex 200. We acknowledge Stephane Rigaud of Image Analysis Hub of C2RI, Institut Pasteur, for his advice, training, and use of workstations to quantification of LAMP1 and Lysotracker. We acknowledge the staff of the breeding zone in the CFJ animal facility (BIME_EOPS) of C2RA, Institut Pasteur, to produce mice for the in vivo experiment. We were thankful to all members of Chromatin and Infection Laboratory and the GORE expertize group of the Hub for their precious advice throughout the project and for critical reading of this manuscript. This project was supported by the Institut Pasteur, the Agence Nationale de la Recherche (ANR-17 CE12 0007 01 EPIBACTIN), the Fondation pour la Recherche Médicale (FRM608 EQU202003010152), the Fondation iXCore-iXLife, the Don Prix CANETTI 2020, the EMBO Young Investigator Program. M.A.H. is a member of the Laboratoire d'Excellence "Integrative Biology of Emerging Infectious Diseases" Agence Nationale de la Recherche (ANR-10-LABX- 62-EIBID). JM was supported by Agence National de la Recherche (ANR-20-PAMR-0011 TheraEPI and by the EUR Genetics and Epigenetics New Education (G.E.N.E.) Graduate school (ANR-17-EURE-0013) which is part of the "Initiative d'Excellence Université de Paris" (ANR-18-IDEX-0001) funded by the French Government through its "Program d'Investissements d'Avenir". MGC was supported by Agence National de la Recherche (ANR LBX-62 IBEID AAP). Bioinformatics and Biostatistics Hub is supported by Institut Pasteur.

## Author contributions

C.Chevalier coordinated the study; C.Chevalier and M.A.H conceived and designed all experiments; C.Chevalier performed experiments and analyzed data; C.Chica supervised all the bioinformatic analysis and performed the transcriptome and integrative analysis; J.M. contributed some experiments and analysis; A.P. performed the bioinformatic analysis for ChIP-seq and MEME-chip. C.Chevalier, C.Chica, and M.A.H. conceived and wrote the manuscript. M.A.H. supervised the work and secured the funding. All authors approved the final manuscript.

## Competing interests

The authors declare no competing interests.
