## [Peer Review File · Nature Communications]

Epithelial cells maintain memory of prior infection with *Streptococcus pneumoniae* through di-methylation of histone H3REVIEWER COMMENTS

Reviewer #1 (Remarks to the Author):

Chevalier and co-workers describe the consequences of epigenetic changes in epithelial cells in response to *Streptococcus pneumoniae*. The authors suggest that much less is known concerning epigenetic changes at mucosal surfaces than in immune cells and it is also true much less is known about how these cells regulate response to pathobionts than immune cells. Furthermore they identify a role for a less characterized dimethylation event H3K4me2 that displays differential kinetics in comparison to the well characterized H3K4me3 event which in their system appears more transient than the H3K4me2 mark which is present for at least 9 days. The use of multiple factor analysis links many H3K4me2 peaks with gene repression and to some of functional changes in endosomal trafficking and metabolism differentially regulated between primary and secondary infections. These are important advances in the field. Further characterization of the relative consequences of the modification would strengthen conclusions and while the impact on internalization, endosomal traffic, metabolism and transcription is clear how this ultimately impacts infection is not completely resolved by the associations identified.

1. The choice of the primary epithelial cell model requires greater justification. What is the consequence for a more differentiated primary epithelial cell type?
2. From Figure 2 The increased in bacterial internalization in secondary infection is associated with lower levels of pHrodo+ bacteria and changes suggesting altered endosomal interactions with vesicles. Are the greater numbers of bacteria a reflection of greater uptake or reduced clearance? Is the reduction in pHrodo events a consequence of overwhelming the endosomal system or of subversion as the authors propose? What are the kinetics of these changes over several time points in the secondary as opposed to primary infections? Is the change in LAMP-1 and the transcriptional changes associated with reduced acidification of the endosome or related to the factors such as NOX2 or inflammasome activation?
3. Further definition of the metabolic changes would be important. These are potentially very important for the response to the bacteria as the authors suggest.
4. Also in Figure 2 there is clear evidence of differential oxidation-reduction and DNA damage between primary and secondary infection. Was DNA damage driven by the oxidation status?
5. What is the turnover or loss of cells following primary infection. Can the authors exclude the possibility that changes are not in cells that survive infection rather that it is selection of traits present at baseline in a heterogenous population at baseline? If you block cell death are representative changes still maintained?
6. For the analysis of the H3K4me2 mark (Figure 3) and related ChIP-seq data (Figure 5) The 3 h time point is used. Some further justification is important. Can the authors also provide representative data at 24 h or other point to exclude any possibility the time point selected has led the team to miss any delayed effects of primary infection? Can some kinetics by immunoblot be provided and quantified for example to study early kinetics of the mark.
7. Some further discussion and any mechanistic insights into the mechanisms of the H3Kme2 mark would be valuable to include.
8. Some further detail on specific response generated by the mark and their consequence to host defence would be valuable. There is already plenty of suggestion from the transcriptional responses and data on endosomal events and metabolism. This would however be important to develop further in particular to support the supposition this is a pathogen subversion of host defence rather than an adaption by the host to regulate inflammatory response?
9. Can the authors exclude any impact of the antibiotic cocktail used that is essential to maintain epithelial cells. While all conditions appear to have had the cocktail evidence that this had no effect on the uninfected sate is advisable to exclude any interaction especially with aminoglycoside antibiotics used here.

Reviewer #3 (Remarks to the Author):

The authors use a mouse model to study epigenetic remodeling of lung epithelial cells in response to bacterial infection. The experiment is interesting and timely, but I found it difficult to follow the RNA-seq and ChIP-seq analysis.

1. Unclear what this means: '... (DEGs, adjusted p-value < 0.05) were determined for the 1° and 2° infections (1° vs UI and 2° vs PI) and between infections ((2° vs PI) vs (1° vs UI)).' How was an adjusted p-value obtained for the 'between infections' comparison? Were the 2nd vs 1st infection groups compared directly or do the authors mean they looked at the overlap between the two initial statistical comparisons?
2. Figure 1D analysis is difficult to follow. Why are downregulated genes combined to make 1,308 (715+593), but upregulated are not? If we look purely at 2nd vs PI, then there are more downregulated genes (750 vs 898). The authors write: '... upregulated DEGs prevail during the 2° infection'. This is not true from the number of genes in that analysis. Second, what does the term 'prevail' mean in the context of gene expression?
3. A simpler comparison is to compare 1st vs 2nd and comment on how different they are from each other.
4. Figure 1E would also be better interpreted as a heatmap or a boxplot. It is hard to know why some are deemed common and some unique to a comparison.
5. For Figure 1F, it would be useful to visualize the metabolism genes and their expression – as a dot plot or similar between groups, to show how much more induced they are in 2nd vs 1st infection.
6. Was there a fold change cut-off applied to identify DEGs? What was the lowest log fold change for a DEG with $p_{adj} < 0.05$?
7. This sentence is confusing: 'These results indicate that cells return to a basal transcriptional state following 1° infection, which is indistinguishable from UI condition.' The '1° infection' is the term used for cells kept in culture and then exposed only once at the same timepoint at PI, UI and 2°. However in this sentence '1° infection' is used to describe the first exposure in the PI sample.
8. The 1° samples are missing from the PCA plot in Figure 1C. Is the top of the figure cut? Only two dots are visible for 2nd infection samples in the bottom right corner.
9. It would be useful for the reader if Figure 1A included the assays that were performed at which timepoint. For example, it is not clear at which timepoint the RNA was collected, until the reader gets to Figure 1D legend.
10. Figure 3A and Figure 4B appear to be contradictory. Is H3K4me2 increased or decreased at 1st infection?
11. The authors use: 'dynamically marked regions (DMRs)'. DMR usually means differentially methylated region in the context of DNA methylation, not histone methylation. May be an unnecessary muddling of terms.
12. It is unclear in Figure 5 and the associated paragraph, what the authors mean by only 12,000 peaks were called in 1st infection compared to PI that had 28,000 peaks. This may indicate a technical issue with the ChIP-sequencing. Were the two sets of samples sequenced to the same depth and show the same quality (e.g. assessed as reads/peaks as a percentage of total reads).
13. It would be great to have some visual representation of the H3K4me2 data. For example a region that corresponds to this statement: 'Interestingly most of the peaks recovered following infection (1°) are shared peaks with both uninfected (UI) and PI conditions, and only ~1 700 (~13%) are unique to that 1° condition, supporting the idea that this time point is transitional.'. For visualization, usually wig files in UCSC Genome browser or similar can be shown to illustrate the point. It is hard to judge the quality of the data and the soundness of the conclusion otherwise.
14. Figure 5C seems to suggest that the strongest H3K4me2 changes are not transient but are maintained between 1st infection and PI.

Reviewer #4 (Remarks to the Author):

S. pneumoniae is an opportunistic pathogen commonly found in the human respiratory tract, where it is a frequent cause of pneumonia. The authors have previously shown that *S. pneumoniae* infection of lung epithelial cells (LECs) results in an altered host transcriptome and epigenome. The current study addresses the question of whether infection alters the ability of those cells to be re-infected, and the mechanistic basis for changes in susceptibility. Using *in vitro* infection of LEC cell lines, the authors find that infection alters the transcriptome of LECs in a heritable way, and that these changes are associated with an increased susceptibility to re-infection. Further, they provide evidence that the transcriptional changes observed are underpinned by increased deposition of the H3K4me2 histone post-translational modification, specifically at transcriptional enhancers.

The data presented in this paper is sound, and the approaches used are appropriate, although it would be helpful if the authors provided examples of ChIP-Seq tracks so that the reader can assess for themselves the quality of the data.

What this paper adds: The authors have previously shown that *S. pneumoniae* infection alters the epigenome and transcriptome of LECs, albeit by different mechanisms to those described here. Further, others have shown that inflammation, including in response to infection, induces long-term transcriptional and epigenomic chromatin changes within skin epithelia that alters responses to secondary injury (<https://doi.org/10.1038/nature24271>), and the concept of induced transcriptional memory in other contexts is not novel. Therefore, the novelty of the current study is the finding that heritable susceptibility to reinfection is dependent on H3K4me2 modulation at particular transcriptional enhancers, not that transcriptional memory occurs, or that memory of infection occurs in non-immune cells. This insight, albeit a very interesting, therefore represents an incremental advance.

What the paper is missing:

- 1) Definitive evidence that H3K4me2 is required for transcriptional memory to occur, or further definitive insights into the mechanism. I understand that the methyltransferase that deposits H3K4me2 also deposits H3K4me3, and so this is not a simple question to address experimentally, but the integrative analysis performed in figure 6 identified transcription factor motifs enriched within the enhancers of genes associated with memory of the infection; presumably the associated TFs are involved in recruitment of the methyltransferase. Could these sites be mutated to abolish methylation? Alternatively, dCAS9-LSD1 fusions have been used to specifically deplete H3K4me2 from targeted genomic regions (e.g. <https://www.ncbi.nlm.nih.gov/pmc/articles/PMC4414811/>).
- 2) Evidence that H3K4me2-mediated transcriptional memory occurs *in vivo* phenomena described occur *in vivo*.

In summary, the authors have presented very well some interesting data, but this study is largely observational, and moreover, based entirely on studies of cell lines. If the authors could some further mechanistic detail, or demonstrate the importance of their findings in an *in vivo* setting, the impact of their study would be increased significantly.

As you will see from the reports copied below, the reviewers raise important concerns. We find that these concerns limit the strength of the study, and therefore we ask you to address them with additional work. Without substantial revisions, we will be unlikely to send the paper back to review. In particular, show that the observed findings also occur in a more physiologically relevant system and provide a more thorough functional analysis of the mark. Please ensure that these and all other concerns raised by our reviewers are addressed in full in a revised manuscript.

We have performed many follow up experiments to answer the reviewers concerns and address point by point each concern in blue below.

REVIEWER COMMENTS

Reviewer #1 (Remarks to the Author):

Chevalier and co-workers describe the consequences of epigenetic changes in epithelial cells in response to *Streptococcus pneumoniae*. The authors suggest that much less is known concerning epigenetic changes at mucosal surfaces than in immune cells and it is also true much less is known about how these cells regulate response to pathobionts than immune cells. Furthermore they identify a role for a less characterized dimethylation event H3K4me2 that displays differential kinetics in comparison to the well characterized H3K4me3 event which in their system appears more transient than the H3K4me2 mark which is present for at least 9 days. The use of multiple factor analysis links many H3K4me2 peaks with gene repression and to some of functional changes in endosomal trafficking and metabolism differentially regulated between primary and secondary infections. These are important advances in the field. Further characterization of the relative consequences of the modification would strengthen conclusions and while the impact on internalization, endosomal traffic, metabolism and transcription is clear how this ultimately impacts infection is not completely resolved by the associations identified.

1. The choice of the primary epithelial cell model requires greater justification. What is the consequence for a more differentiated primary epithelial cell type?

We have addressed this point by performing *in vivo* experiments. We performed intranasal mouse infections, physiologically relevant for *S. pneumoniae*, and used FACS analysis to measure H3K4me2 levels specifically in lung epithelial cells. We have added these results to figure 3 panels E and F and show that the increase in methylation is recapitulated in this physiological model, which includes both undifferentiated and differentiated primary epithelial cells.

2. From Figure 2 The increased in bacterial internalization in secondary infection is associated with lower levels of pHrodo+ bacteria and changes suggesting altered endosomal interactions with vesicles. Are the greater numbers of bacteria a reflection of greater uptake or reduced clearance?

We would like to clarify for the reviewer that we have not used pHrodo in our manuscript and are not measuring internalization of bacteria as *S. pneumoniae* is mostly extracellular. Our data indicate that the greater number of CFU recovered following secondary infection are adhered bacteria and not internalized bacteria. We have made this point clearer in the revised version of the manuscript, and we have added data in figure 2A and B and S2B supporting this point. Therefore, we argue that the greater number of bacteria recovered after secondary infection reflects greater adhesion.

Is the reduction in pHrodo events a consequence of overwhelming the endosomal system or of subversion as the authors propose?

We have performed experiments to further study the endosomal system and show this in figure S2B. Here, we provide immunofluorescence images during primary and secondary infections of LAMP1 and bacterial staining. We show that both stainings are not colocalized demonstrating that infection is not “overwhelming the endosomal system”, and in fact, the changes in the endosomal system are not due to internalization of bacteria. We have also added data supporting greater adherence of bacteria upon secondary infection (Figure 2A) and that adherence of bacteria (Figure S4E) (and not internalization) is necessary for H3K4me2 (Figure 4F).

What are the kinetics of these changes over several time points in the secondary as opposed to primary infections?

If we understood the reviewer's request and it pertaining to endosomal trafficking, we have now added another time point for the study of endosomal trafficking in Figure 2D. We now have 3h and 24h post primary and secondary infection.

Is the change in LAMP-1 and the transcriptional changes associated with reduced acidification of the endosome or related to the factors such as NOX2 or inflammasome activation?

We have added a more detailed study of the endosomal trafficking following primary and secondary infection in figure 2D and S2A. With these careful quantifications we demonstrate that LAMP1 levels increase 3h post infection, and to a higher level upon secondary infection. More strikingly are the differences in lysosomal acidification as measured by LysoTracker. With this marker we clearly show that secondary infection strongly increases the acidity of lysosomes compared to primary infection.

3. Further definition of the metabolic changes would be important. These are potentially very important for the response to the bacteria as the authors suggest.

We have now added the individual gene list in the manuscript under the following link:

Primary infection:

https://gitlab.pasteur.fr/cchica/epimem/-/blob/main/results/FGSEA_REACTOME_H3vsUI.txt?ref_type=heads

Secondary infection:

https://gitlab.pasteur.fr/cchica/epimem/-/blob/main/results/FGSEA_REACTOME_DIvsD7.txt?ref_type=heads

4. Also in Figure 2 there is clear evidence of differential oxidation-reduction and DNA damage between primary and secondary infection. Was DNA damage driven by the oxidation status?

In fact, we do not see differences in DNA damage between first and second infections. We have clarified this information in the text and placed DNA damage results in the supplementary figure S2C as these data show no change in gH2AX levels. Therefore, the changes in oxidation status are uncorrelated from DNA damage.

5. What is the turnover or loss of cells following primary infection. Can the authors exclude the possibility that changes are not in cells that survive infection rather that it is selection of traits present at baseline in a heterogenous population at baseline? If you block cell death are representative changes still maintained?

We have performed complimentary experiments to answer this point and have included the results in Figure S1A. Here we stained cells with viability markers Calcein AM (cell permeable dye metabolized and retained only in live cells) and DRAQ7 (cell impermeable DNA dye that will only stain cells with membrane integrity issues) and show that more than 94% cells are alive after 3h of infection. Therefore, we are not selecting a subpopulation of cells in our assay as the large majority of them (over 94%) survive infection and are carried through in our assay. In combination with the CFSE staining shown in figure S1B, we can conclusively say that cells survive and divide following infection.

6. For the analysis of the H3K4me2 mark (Figure 3) and related ChIP-seq data (Figure 5) The 3 h time point is used. Some further justification is important. Can the authors also provide representative data at 24 h or other point to exclude any possibility the time point selected has led the team to miss any delayed effects of primary infection? Can some kinetics by immunoblot be provided and quantified for example to study early kinetics of the mark.

We would like to clarify the timing of our experiments. Western blots and ChIP-seq were acquired 3h post infection and then 7 days post infection. However, immunofluorescence experiments were performed at 48h post infection as this time point reliably and reproducibly shows the increase in H3K4me2 (methylation is not reliably measured using this method before 48h). We have now added an earlier timepoint by immunofluorescence, 24h in figure 4C. Therefore, with all experiments combined, we have provided a time course of this mark spanning from 3h, to 24h, to 48h, to 9 days, which are all in agreement.

7. Some further discussion and any mechanistic insights into the mechanisms of the H3Kme2 mark would be valuable to include.

We have performed many experiments to address this point both on the bacterial and the cellular side of the mechanisms. Here are the main points:

Bacterial side: - We have performed experiments with mutant bacteria to show that the increase in H3K4me2 is capsule independent but pilus dependent (results shown in figure 4E and F). Together with the bacterial counts for these mutants (Fig S4D and E), we can now state that bacterial adhesion is driving the increase in H3K4me2.

- We have performed IF experiments and can show that bacteria are adhered on the outside of cells, confirming that bacterial adhesion is driving the modification and not internalization (Figure S2B).

Cellular side: - We have measured levels of mono-methylation and show that only di-methylation (we had already shown the results for tri-methylation) is maintained over time and therefore this mark is specific. These results are included in a new figure 5 A and B.

- We have performed experiments in the presence of a general methyltransferase inhibitor Sinefungin. These results are included in figure 5C, D, E and F and show that even in the presence of inhibitor, di-methylation still occurs. Therefore, these data suggest that we can rule out that a methyltransferase is involved. In the discussion we propose the hypothesis that infection could be blocking a demethylase instead, a mechanism that would be more complicated to test for as inhibition would not address this point.

- We have performed experiments to show that the increase in H3K4me2 is independent of another histone modification that *S. pneumoniae* induces, H3S10 dephosphorylation (Figure 5G and H). These data show that H3 dephosphorylation is not a precursor to H3 di-methylation.

8. Some further detail on specific response generated by the mark and their consequence to host defence would be valuable. There is already plenty of suggestion from the transcriptional responses and data on endosomal events and metabolism. This would however be important to develop further in particular to support the supposition this is a pathogen subversion of host defence rather than an adaption by the host to regulate inflammatory response?

We have added data in figure S1D showing a multiplex ELISA of host responses to infection following first and second infections. These results show that there are no significant differences in the levels of any of these, even

though there are 10 times more bacteria upon secondary infection. We therefore conclude that the host is not adapting to the presence of bacteria and increasing defense mechanisms. In this new manuscript we have also added data showing that bacterial adherence is increased upon secondary infection, further demonstrating an advantage for bacteria rather than cellular defense.

9. Can the authors exclude any impact of the antibiotic cocktail used that is essential to maintain epithelial cells. While all conditions appear to have had the cocktail evidence that this had no effect on the uninfected state is advisable to exclude any interaction especially with aminoglycoside antibiotics used here. Please find below the results comparing cell treated versus untreated with antibiotics for the same time as performed in the infection conditions (48h). No significant differences were found over 3 experiments, and therefore we have chosen not to include this data in the manuscript.

Reviewer #3 (Remarks to the Author):

The authors use a mouse model to study epigenetic remodeling of lung epithelial cells in response to bacterial infection. The experiment is interesting and timely, but I found it difficult to follow the RNA-seq and ChIP-seq analysis.

We would like to clarify that all the figures in our first manuscript were performed using in vitro tissue cultured cells, and not in vivo mouse studies. Please find below clarifications on RNA-seq and ChIP-seq analyses.

1. Unclear what this means: '... (DEGs, adjusted p-value < 0.05) were determined for the 1° and 2° infections (1° vs UI and 2° vs PI) and between infections ((2° vs PI) vs (1° vs UI)).' How was an adjusted p-value obtained for the 'between infections' comparison? Were the 2nd vs 1st infection groups compared directly or do the authors mean they looked at the overlap between the two initial statistical comparisons?

For the "between infections comparison", we did a direct comparison using a specific contrast. To clarify this point, we added the following text to the "Transcriptome analysis" paragraph in the **Methods** section. We wrote:

Considering the four conditions: UI, 1°, PI and 2° we tested the following contrasts:

- primary infection: 1° - UI
- secondary infection: 2° - PI
- between infections: ((2° - PI) - (1° - UI)).

Genes with an adjusted p-value < 0.05, for the first two contrast, or an adjusted p-value < 0.3, for the last one, were defined as differentially expressed (DEGs). The two first comparisons were used in the general characterization of the transcriptome, Figure 1D. The last comparison, which includes genes whose change in the 2° infection in comparison to PI is significantly bigger/smaller than the change in the 1° infection in comparison to UI, was used in the Multiple Factor Analysis (MFA) to classify gene changes as Only 1°, Only 2° or Not specific (Figure 7).

2. Figure 1D analysis is difficult to follow. Why are downregulated genes combined to make 1,308 (715+593), but upregulated are not? If we look purely at 2nd vs PI, then there are more downregulated genes (750 vs 898). The authors write: '... upregulated DEGs prevail during the 2° infection'. This is not true from the number of genes in that analysis. Second, what does the term 'prevail' mean in the context of gene expression?

Following the reviewer's comment, we have simplified the global description of the transcriptome. For Figure 1C we consider now only the primary and secondary comparisons, i.e. 1° - UI and 2° - PI. We have adapted the text accordingly.

3. A simpler comparison is to compare 1st vs 2nd and comment on how different they are from each other.

We chose to compare each infection to the corresponding "steady state", that is UI for 1° and PI for 2° infection. This because, even if the primary source of variability of the time course (PC1 ~50%) coincides with the differences between UI/PI time points and the 1° and 2° infections, there is still some variability between the two steady points, UI and PI (PC3 ~9.6%), comparable to the variability between infections (PC2 ~11%). The comparison with the corresponding steady state guarantees that transcriptomic changes between IU and PI are considered when comparing to the corresponding infection.

We have added the second PCA to Figure 1B and the above text to the Methods section description.

4. Figure 1E would also be better interpreted as a heatmap or a boxplot. It is hard to know why some are deemed common and some unique to a comparison.

Gene expression changes are indeed similar between 1° and 2° infection for most pathways. However, those defined as “specific” to 1° or 2° infection show stronger changes in one infection with respect to the other. This corresponds to a FDR < 0.05 and a bigger absolute normalized enrichment score (NES), as obtained from the gene set enrichment analysis (GSEA). Conversely, pathways defined as common to both infections are significantly enriched for both infections (FDR < 0.05).

5. For Figure 1F, it would be useful to visualize the metabolism genes and their expression – as a dot plot or similar between groups, to show how much more induced they are in 2nd vs 1st infection.

To answer to this point and to further clarify the above point 4, we followed the reviewer’s suggestion and added in the Supplementary Figure 1C plots comparing the log fold change for the 2° vs the 1° infections.

6. Was there a fold change cut-off applied to identify DEGs? What was the lowest log fold change for a DEG with padj < 0.05?

There was log fold threshold applied to define the DEGs. Below the distribution of the corresponding log fold changes for each comparison.

7. This sentence is confusing: ‘These results indicate that cells return to a basal transcriptional state following 1° infection, which is indistinguishable from UI condition.’ The ‘1° infection’ is the term used for cells kept in culture and then exposed only once at the same timepoint at PI, UI and 2°. However, in this sentence ‘1° infection’ is used to describe the first exposure in the PI sample.

We thank the reviewer for this comment and have changed the sentence in the text to clarify this point.

8. The 1° samples are missing from the PCA plot in Figure 1C. Is the top of the figure cut? Only two dots are visible for 2nd infection samples in the bottom right corner.

We are very sorry and realize there was a problem with our figure which had a white panel hiding the data. We have now provided the correct figure without the hidden points for better understanding.

9. It would be useful for the reader if Figure 1A included the assays that were performed at which timepoint. For example, it is not clear at which timepoint the RNA was collected, until the reader gets to Figure 1D legend.

We thank the reviewer for this suggestion and have clarified the experimental procedure in figure 1A. We now indicate 3 points at which samples are collected throughout the manuscript (α , β , γ). In each figure legend it is now indicated at which point samples were collected.

10. Figure 3A and Figure 4B appear to be contradictory. Is H3K4me2 increased or decreased at 1st infection?

At the timepoint directly following first infection (3h) the levels of H3K4me2 are not yet detectable, as shown in figure 3A. Figure 4B detects levels 48h post infection, at which point H3K4me2 is reliably detectable, and these levels remain high for at least 9 days (figure 4B shows 7 days). Therefore, we believe this was a misunderstanding of the time points at which data was collected rather than a contradiction of figures. We have now added clarification points for the reader to better understand when samples were collected.

11. The authors use: ‘dynamically marked regions (DMRs)’. DMR usually means differentially methylated region in the context of DNA methylation, not histone methylation. May be an unnecessary muddling of terms.

We are sorry this is confusing for the reviewer, but as there is no DNA methylation in this manuscript, we believe a definition for the abbreviation should be sufficient.

12. It is unclear in Figure 5 and the associated paragraph, what the authors mean by only 12,000 peaks were called in 1st infection compared to PI that had 28,000 peaks. This may indicate a technical issue with the ChIP-

sequencing. Were the two sets of samples sequenced to the same depth and show the same quality (e.g. assessed as reads/peaks as a percentage of total reads).

QC metrics of all the ChIPseq libraries have been added as supplementary material and a paragraph clarifying this point added in the Methods' section. Libraries have comparable sequencing depth and quality.

The main reason for the low number of reproducible peaks for the 1° infection is the higher variation between replicates, which can be appreciated in the PCA Figure 6C. As explained in the **Methylome analysis paragraph of the Methods** section, reproducible peaks per condition were determined using the intersection approach, i.e. requiring at least 40% length overlap between replicates. For the first description of the methylome shown in Figure 6A, we consider the intersection between reproducible peaks among samples, which is of course dependent on the number of peaks initially called. However, the differential analysis and subsequent analysis done to study the methylation dynamics along 1° infection we used the non-redundant set of reproducible peaks merged among **all** conditions.

A sentence clarifying this last point was added to the **Methylome analysis paragraph of the Methods** section. Additionally, the section **Methylome dynamics during 1° infection** was re-worded to better convey the message.

13. It would be great to have some visual representation of the H3K4me2 data. For example a region that corresponds to this statement: 'Interestingly most of the peaks recovered following infection (1°) are shared peaks with both uninfected (UI) and PI conditions, and only ~1 700 (~13%) are unique to that 1° condition, supporting the idea that this time point is transitional.'. For visualization, usually wig files in UCSC Genome browser or similar can be shown to illustrate the point. It is hard to judge the quality of the data and the soundness of the conclusion otherwise.

Genome browser examples for each peak category were added, Fig 6E. The corresponding bigwig files were also added to the GEO dataset.

14. Figure 5C seems to suggest that the strongest H3K4me2 changes are not transient but are maintained between 1st infection and PI.

We agree with this statement, and this is the main point and novelty of the manuscript.

Reviewer #4 (Remarks to the Author):

S. pneumoniae is an opportunistic pathogen commonly found in the human respiratory tract, where it is a frequent cause of pneumonia. The authors have previously shown that S. pneumoniae infection of lung epithelial cells (LECs) results in an altered host transcriptome and epigenome. The current study addresses the question of whether infection alters the ability of those cells to be re-infected, and the mechanistic basis for changes in susceptibility. Using in vitro infection of LEC cell lines, the authors find that infection alters the transcriptome of LECs in a heritable way, and that these changes are associated with an increased susceptibility to re-infection. Further, they provide evidence that the transcriptional changes observed are underpinned by increased deposition of the H3K4me2 histone post-translational modification, specifically at transcriptional enhancers.

The data presented in this paper is sound, and the approaches used are appropriate, although it would be helpful if the authors provided examples of ChIP-Seq tracks so that the reader can assess for themselves the quality of the data.

Genome browser examples for each peak category were added, Fig 6E. The corresponding bigwig files were also added to the GEO dataset. Furthermore, QC metrics of all the ChIPseq libraries have been added as supplementary material for the reader to assess the quality of the data.

What this paper adds: The authors have previously shown that S. pneumoniae infection alters the epigenome and transcriptome of LECs, albeit by different mechanisms to those described here. Further, others have shown that inflammation, including in response to infection, induces long-term transcriptional and epigenomic chromatin changes within skin epithelia that alters responses to secondary injury (<https://doi.org/10.1038/nature24271>), and the concept of induced transcriptional memory in other contexts is not novel. Therefore, the novelty of the current study is the finding that heritable susceptibility to reinfection is dependent on H3K4me2 modulation at particular transcriptional enhancers, not that transcriptional memory occurs, or that memory of infection occurs in non-immune cells. This insight, albeit a very interesting, therefore represents an incremental advance.

We agree with the reviewer that the cited paper (<https://doi.org/10.1038/nature24271>) was pioneering in showing that epithelial cells maintain transcriptional memory and epigenetic changes for the first time. We kindly disagree that this publication includes infection data, as in this paper the stimulus was inflammation induced by imiquimod. To our knowledge bacterial infection has not been shown to leave a lasting imprint in epithelial cells, and our manuscript is novel for that. We further agree that innate immune memory is not a novel concept, but our manuscript shows that the response is actively mediated by bacteria and is therefore different from innate immune memory. In our updated manuscript we have strengthened this point and show that cells do not have a heightened antibacterial response and in contrast are more permissive to bacterial adhesion upon secondary infection, which is the opposite phenotype than innate immune memory, and further bringing novelty to our work.

What the paper is missing:

1) Definitive evidence that H3K4me2 is required for transcriptional memory to occur, or further definitive insights into the mechanism. I understand that the methyltransferase that deposits H3K4me2 also deposits H3K4me3, and so this is not a simple question to address experimentally, but the integrative analysis performed in figure 6 identified transcription factor motifs enriched within the enhancers of genes associated with memory of the infection; presumably the associated TFs are involved in recruitment of the methyltransferase.

Could these sites be mutated to abolish methylation? Alternatively, dCAS9-LSD1 fusions have been used to specifically deplete H3K4me2 from targeted genomic regions (e.g. <https://www.ncbi.nlm.nih.gov/pmc/articles/PMC4414811/>).

We agree with the reviewer that providing definitive evidence that H3K4me2 is required for transcriptional memory is not a simple question to address experimentally, but we have tried our best. Indeed, as the reviewer mentions, there are more than 10 known methyltransferases, and they all target both di-methylation and tri-methylation. However, we have tried and performed several different experiments to address this point.

1) We have used a general inhibitor to the AP-1 transcription factor family (SR11302 inhibitor), which we have found to be the main TF whose binding motif correlates with positions of H3K4me2 peaks. We reasoned that if this TF family was important for localizing H3K4me2 peaks, which we understand is suggested by the reviewer, we would see a difference in di-methylation levels. The experimental procedure is shown below as well as the results.

We first controlled for inhibitor activity by performing WB against phosphorylated c-jun (which should be a target of the SR11302 inhibitor). Unfortunately, for reasons we cannot explain the inhibitor had the opposite effect than expected on phospho c-jun as it massively increased the levels in both uninfected and infected conditions (see middle graph below). This large increase in phospho c-jun correlated with an increase in the basal level of H3K4me2, even without infection. Although we no longer observe an additional increase in H3K4me2 upon infection, it is hard to conclude as the basal levels are so greatly modified. Therefore, we are not confident that the block in methylation upon infection is due to the inhibitor blocking AP-1, and these results are inconclusive. Given the impossibility to interpret these results we are showing them for the reviewer but will not be including them in the paper.

2) To definitely demonstrate that H3K4me2 is necessary for transcriptional memory or phenotypes associated with second infection, we need to be able to block this modification. We have included new data in the manuscript using a pan methyltransferase inhibitor, sinefungin (figure 5C-F). Here, we show that Sinefungin, although effective in blocking H3K4me1 levels, does not block infection-induced H3K4me2. These results suggest that infection is not modulating methyltransferases, rather other host machinery. These data further the mechanistic understanding of the phenomenon but do not provide definitive evidence that H3K4me2 is required for transcriptional memory.

3) To provide evidence that H3K4me2 is required for the phenotype on second infection, we have added data in figure 2B. Here, we have performed 1° infection with inactivated bacteria, which we show in figure 4B do not induce H3K4me2. We then evaluated the phenotype of 2° infection for the CFUs recovered. In this experiment we show that the increase in CFU that we observe in 2° infection following live 1° infection, is lost. These data are in agreement with the modifications in metabolic activity, which are also dependent on H3K4me2 being modified. These data therefore indicate that H3K4me2 is necessary to induce the change in phenotype observed in 2° infection.

4) We have significantly advanced the mechanism of how bacteria mediate H3K4me2 in figure 4E and F, by including data using different bacterial mutants and show that adhesion is required.

2) Evidence that H3K4me2-mediated transcriptional memory occurs in vivo phenomena described occur in vivo. We have now added an *in vivo* experiment in figure 3E and F that shows that in a physiological model of intranasal infection, H3K4me2 levels are increased in epithelial cells 3 days post infection.

In summary, the authors have presented very well some interesting data, but this study is largely observational, and moreover, based entirely on studies of cell lines. If the authors could some further mechanistic detail, or demonstrate the importance of their findings in an in vivo setting, the impact of their study would be increased significantly.

We thank the reviewer and have now added a large amount of new data, which we believe significantly increases the impact of this study and directly addresses this reviewer, as well as the others' concerns.

REVIEWER COMMENTS

Reviewer #1 (Remarks to the Author):

The authors appear to have responded to the issues I have raised.

Reviewer #3 (Remarks to the Author):

I thank the authors for addressing my comments. There are still a couple of points to complete.

1. The authors still use the term 'prevail', to try to say that 2nd infection is trained and different to 1st infection: 'Indeed, earlier in the differential expression analysis, we highlighted that after 1^o infection, cells respond by downregulated DEGs while upregulated DEGs prevail in the 2^o infection.' What does 'prevail' mean in this context? In PCA plot Figure 1B, we can see that 1st and 2nd infection both cluster to the right of PC1 (this PC explains 47%). Therefore their overall response is similar. Also in figure 1C: 1st v UI (927 up, 1001 down genes) for 2nd v PI (750 up 898 down), we see that in both cases there more downregulated genes

2. The authors say they added a new PCA to Figure 1B, but there still only the original. This time fixed with all samples shown. Is there supposed to be a second PCA showing separation of UI and PI on PC3?

3. Figure 1E should be a heatmap to highlight the extent to which the 'specific to 1st' genes really are. For example, cluster 10 in the first panel of Figure 1E does not look specific to 1st infection because the circle and triangle are touching. It may indeed be a simple cut-off issue, with one being 0.049 and one being 0.051, but the color of the heatmap would show us that the effects are probably the same, and there is nothing 'unique' about the overall response.

4. The authors have explained that they used a logFC cut-off to identify DEGs, thank you. This should also be added to the methods section of the paper.

Reviewer #4 (Remarks to the Author):

The authors have largely addressed the concerns that I had about the original manuscript. I feel that the authors' claims of the mechanistic basis for, and the physiological relevance of their findings are now strengthened considerably, and I agree with them that the likely impact of the manuscript has been significantly increased. My suggestion is that the revised manuscript be published as is.

Reviewer #3 (Remarks to the Author):

I thank the authors for addressing my comments. There are still a couple of points to complete.

1. The authors still use the term 'prevail', to try to say that 2nd infection is trained and different to 1st infection: 'Indeed, earlier in the differential expression analysis, we highlighted that after 1° infection, cells respond by downregulated DEGs while upregulated DEGs prevail in the 2° infection.' What does 'prevail' mean in this context? In PCA plot Figure 1B, we can see that 1st and 2nd infection both cluster to the right of PC1 (this PC explains 47%). Therefore their overall response is similar. Also in figure 1C: 1st v UI (927 up, 1001 down genes) for 2nd v PI (750 up 898 down), we see that in both cases there more downregulated genes

We agree with the reviewer that the sentence including the term "prevail" is confusing and therefore we have now removed it from the revised manuscript.

2. The authors say they added a new PCA to Figure 1B, but there still only the original. This time fixed with all samples shown. Is there supposed to be a second PCA showing separation of UI and PI on PC3?

We are sorry for the confusion. The point the reviewer is making is in regard to our response to his/her point 3. Here we pointed out that we needed to compare UI to 1° infection, and PI to 2° infection, due to differences observed between UI and PI, independently of infection. These differences can be observed in PC3 shown below, which we choose to show the reviewer for transparency but don't consider necessary to add to the manuscript. Indeed, biologically, samples need to be compared to their untreated control, which we do not find necessary to justify (and complicate) by showing a PCA with another variable.

3. Figure 1E should be a heatmap to highlight the extent to which the 'specific to 1st' genes really are. For example, cluster 10 in the first panel of Figure 1E does not look specific to 1st infection because the circle and triangle are touching. It may indeed be a simple cut-off issue, with one being 0.049 and one being 0.051, but the color of the heatmap would show us that the effects are probably the same, and there is nothing 'unique' about the overall response.

We have now converted Figure 1E to a heatmap representation, which highlights well the specificity in the responses versus the common responses, and does not change the message on the overall infection specificity (or not) of the biological pathways. We would like to point out that the expression profile of the two infections is not expected to be completely antagonistic as cells are still responding to the same strong stimulus. However, we still observe differences that are more or less subtle depending on the functional category. In addition, we have now removed the Pathway Immune system, which is the one the reviewer refers to and had the smallest difference between 1° and 2° infection.

4. The authors have explained that they used a logFC cut-off to identify DEGs, thank you. This should also be added to the methods section of the paper.

We did not used any log fold threshold to identify DEGs. We apologize if the message was not clear in our first reply. The log fold distributions correspond to the log folds of DEGs selected using only an adjusted p-value threshold.

REVIEWERS' COMMENTS

Reviewer #3 (Remarks to the Author):

The authors have addressed all my comments, thank you.